# Long term carbon export from mountain forests driven by hydroclimate and extreme event driven landsliding

Jamie D. Howarth [1,2] ✉, Sean J. Fitzsimons [3], Adelaine Moody[1], Jin Wang[4,5], Mark H. Garnett [6], Thomas Croissant[4], Alex L. Densmore[4], Andy Howell[7,8] & Robert G. Hilton [2]

The export of organic carbon from terrestrial ecosystems by erosion may play a central role in balancing the geological carbon cycle and Earth's climate over millennial timescales. However, constraints on organic carbon yields have come from sampling modern rivers that don't capture variation over decades to millennia driven by changing hydroclimate and erosion during extreme events. Here we use volumetric reconstructions of lake sedimentary fills to generate timeseries of sediment and organic carbon yields from two catchments draining the Southern Alps, New Zealand over the last millennium. The reconstructed yields indicate that earthquake-induced landslides significantly increase sediment and organic carbon yields, contributing to pulsed export that accounts for ~40% of the total. Between extreme events, organic carbon export increased twofold during centuries with a wetter reconstructed climate. Our findings suggest that the link between hydroclimate and organic carbon export may act as a negative feedback in the longer-term carbon cycle.

The geological carbon cycle is characterized by carbon fluxes between atmospheric, biospheric and lithospheric reservoirs that mediate atmospheric $CO_2$ and global climate over millennial to million-year timescales[1,2]. In canonical views of this carbon cycle, weathering of silicate minerals freshly exposed at Earth's surface by uplift and erosion of mountains draws down $CO_2$ from the atmosphere, balancing the emissions from Solid-Earth degassing, and keeping global climate in a habitable state[2,3]. However, by quantifying the role of other weathering reactions, a growing body of work suggests weathering may be a net source of $CO_2$ to the atmosphere because the net drawdown of $CO_2$ from silicate weathering (47–72 Mt C yr$^{-1}$ globally after accounting for carbonate mineral formation)[4] is offset or even exceeded by $CO_2$ emissions from oxidative weathering of petrogenic organic carbon ($OC_{petro}$; 62–86 Mt C yr$^{-1}$)[5], and sulfide minerals (30–40 Mt C yr$^{-1}$)[6]. Therefore, to maintain a habitable climate over the Phanerozoic, a large additional sink of $CO_2$ is required, which re-emphasises the role of organic carbon burial[2,7].

Erosion of the continents contributes to organic carbon burial by mobilizing and transporting biospheric organic carbon ($OC_{bio}$) from vegetation and soils to the ocean (107–231 Mt C yr$^{-1}$ globally; henceforth referred to as '$OC_{bio}$ export'))[7–9], and by increasing mass accumulation rates on continental margins enhancing $OC_{bio}$ burial efficiency[10]. Thus, quantifying $OC_{bio}$ yields from terrestrial ecosystems and identifying the processes that control them over millennial timescales is essential for understanding the global carbon cycle[2]. However, almost all previous estimates of $OC_{bio}$ export from terrestrial environments and inferences about the processes that control it have been derived from river gauging of suspended sediment (SS) and carbon yields that span years to a decade at most[8,9]. This is a major challenge because key carbon cycle, geomorphic and sedimentary processes play out over millennia and may have periods of adjustment to extreme events[11] or climate change[12] over decades to millennia, which cannot be assessed from river gauging records over timeframes capturing less than a decade.

Temporal variability in $OC_{bio}$ export from terrestrial ecosystems to the ocean may occur due to pulsed perturbations (e.g. rapid changes in magnitude of fluxes over timescales of years to decades) or more gradual change (e.g. secular change over timescales of 100–1000 s of years). Despite their rarity, large earthquakes and storms significantly perturb Earth surface dynamics by triggering landslides that mobilize sediment and carbon on hillslopes and increase loads in rivers[13–15]. River suspended sediment yields can increase by up to an order of magnitude and persist for years to decades

¹School of Geography, Environment and Earth Sciences, Te Herenga Waka - Victoria University of Wellington, Wellington, New Zealand. ²Department of Earth Sciences, University of Oxford, Oxford, UK. ³School of Geography, University of Otago, Dunedin, New Zealand. ⁴Department of Geography, Durham University, Durham, UK. ⁵State Key Laboratory of Loess and Quaternary Geology, Institute of Earth Environment, Chinese Academy of Sciences, Xi'an, China. ⁶NEIF Radiocarbon Laboratory, East Kilbride, UK. ⁷Department of Earth Structure and Processes, GNS Science, Lower Hutt, New Zealand. ⁸School of Earth and Environment, University of Canterbury, Christchurch, New Zealand. ✉e-mail: Jamie.Howarth@vuw.ac.nz

after the triggering events[11,13–15], but comparatively little is known about how earthquake- and storm-induced landsliding contributes to long-term $OC_{bio}$ export from terrestrial ecosystems due to the brevity of most records[16–19]. In addition, secular changes in climate (e.g. late Holocene climate change) could impact runoff and associated $OC_{bio}$ export[9], yet we lack assessments of how yields have changed over these longer periods of time.

Relatively short records of riverine $OC_{bio}$ yields have also precluded direct assessment of the controls on $OC_{bio}$ export variability through time. The relationship between $OC_{bio}$ yields, SS yields (as a proxy for physical erosion) and runoff from both global[8] and mountain[9] rivers has been revealed using spatial correlations between these variables. These correlations suggest the rate of $OC_{bio}$ export scales positively with physical erosion[8] and runoff[9] implying there may be feedbacks between OC export by rivers, tectonic processes and climatic change that could regulate net $CO_2$ transfers through space and time[2]. However, implicit in this interpretation is the assumption that apparent spatial correlations between variables inform controls on temporal variability, a "space for time" substitution[20,21]. This assumption may not be valid, especially in cases where scatter in the scaling between $OC_{bio}$ yields and runoff spans an order of magnitude, potentially obscuring multiple inter-related controls. These assumptions can only be tested with records of $OC_{bio}$ export that span decades to millennia. It is clear that longer term records of $OC_{bio}$ export that span decades to millennia are required to refine our understanding of how $OC_{bio}$ export mediates the global carbon cycle.

Here we use an alternative approach to global datasets of river yields: we keep the physiographic setting constant while reconstructing past sediment and carbon yields from catchments using the sediment record. A high spatial density coring approach reconstructs mass accumulation in two lakes basins[22], allowing SS, $OC_{bio}$ and $OC_{petro}$ export to be quantified from two ($\sim 60\ km^2$) range front catchments of the Southern Alps, New Zealand over the last millennium (Fig. 1). During this time there have been four $M_w > 7.9$ earthquakes on the range-bounding Alpine Fault (recurrence interval: $251 \pm 48$ ($1\sigma$) years)[23], storms that have resulted in widespread slope instability[24], and late Holocene changes in hydroclimate[25]. We also consider these findings in the context of global rates of earthquake-triggered landsliding, and in the context of models of $OC_{bio}$ export, which suggest that hydroclimate moderates carbon export from mountain forests. By combing observations from instrumental and natural archives that span years to a millennium, we establish links between tectonic and climatic forcing and carbon export from forests by quantifying how: 1) earthquakes and storms drive pulsed increases in sediment and $OC_{bio}$ export over decades; and 2) precipitation variability due to climatic change sets longer-term patterns in

OC export from terrestrial ecosystems in the western range front of the Southern Alps.

## Results and discussion
### Reconstructed sediment and carbon yields

The sedimentary records of lakes Paringa and Mapourika are characterized by a repeating sequence of sedimentary phases that can be related to the Alpine Fault earthquake cycle[11,24,26] (Fig. 2a, d): (i) co-seismic megaturbidites formed by shaking-induced subaqueous mass wasting (henceforth termed co-seismic phase); (ii) post-seismic hyperpycnite stacks formed during periods of elevated river sediment flux in response to earthquake-induced landsliding (henceforth termed post-seismic phase); and (iii) inter-seismic layered silts formed between earthquakes when the catchments are responding to secular changes in environmental forcing (henceforth termed inter-seismic phase). Within some inter-seismic phases there were isolated hyperpycnite stacks inferred to be associated with storm-driven landsliding[24]. We used a dense network of ~6 m sediment cores from the lakes to establish detailed stratigraphic correlations that formed the basis for volumetric and density models of the sedimentary fill (Methods; Fig. 2; Supplementary Figs. 1–3; Supplementary Data 1).

Combined the volumetric and density models for Lake Paringa show that $6.7 \times 10^6 \pm 0.2 \times 10^6$ t (2 Standard Errors (SE)) of sediment accumulated over the last 986 years (965–1007 years; 95% Highest Probability Density Function (HPDF) range). $6 \pm 0.4\%$ (2SE) of the total mass was partitioned into co-seismic phases, $39 \pm 3.0\%$ (2SE) into post-seismic phases and $55 \pm 3.0\%$ (2SE) into inter-seismic phases. Equivalent models for Lake Mapourika have $6.6 \times 10^6 \pm 0.3 \times 10^6$ t (2SE) of sediment over the last 553 years (547–549 years; 95% HPDF range), $17 \pm 1.0\%$ (2SE) in co-seismic phases, $34 \pm 3.0\%$ (2SE) in post-seismic phases and $38 \pm 4.0\%$ (2SE) in inter-seismic phases. Two hyperpycnite stacks related to storm-induced land-sliding within the inter-seismic phases make up the remaining volume.

Dry sediment masses combined with measurements of the proportion of organic carbon in these phases and high resolution age-depth models (Methods; Supplementary Figs. 6–8) allowed SS, $OC_{bio}$ and $OC_{petro}$ yields to be quantified (Fig. 3a). Coseismic deposits were formed by reworking of lacustrine sediment on deltas and subaqueous slopes and do not relate directly to catchment yields through time so are omitted[11]. In Lake Paringa the average SS and $OC_{bio}$ yields during the post-seismic phases were significantly higher than the average inter-seismic yields (SS yields: $\chi^2 = 40.1$, df = 1, $p < 0.01$; $OC_{bio}$ yields: $\chi^2 = 27.5$, df = 1, $p < 0.01$). Inter-seismic SS and $OC_{bio}$ yields also varied significantly between earthquake cycles (SS yields:

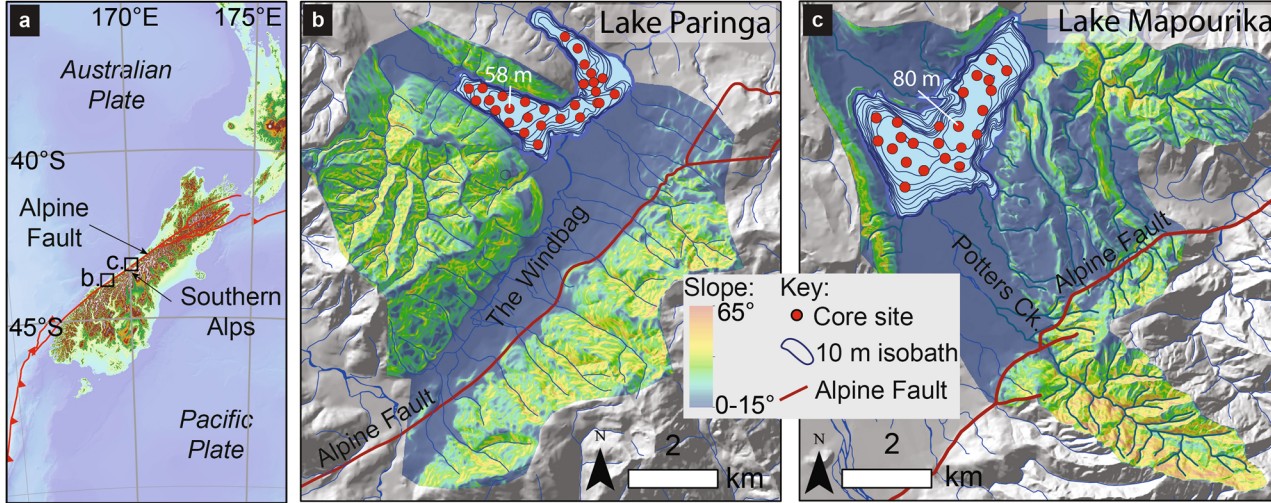

**Fig. 1 | Tectonic settings and catchment maps for lakes Paringa and Mapourika. a** The tectonic setting and topography of the Southern Alps. **b, c** The rangefront catchments of lakes Paringa and Mapourika. Coloured areas show lake catchments, and colour range shows hillslope angle thresholded to >15° to depict slopes from which most sediment is sourced[26]. The red line is the surface trace of the range-bounding Alpine Fault, blue polygons in the lakes represent 10 m isobaths, and red circles show the location of core sites used to reconstruct sedimentary fill volumes.

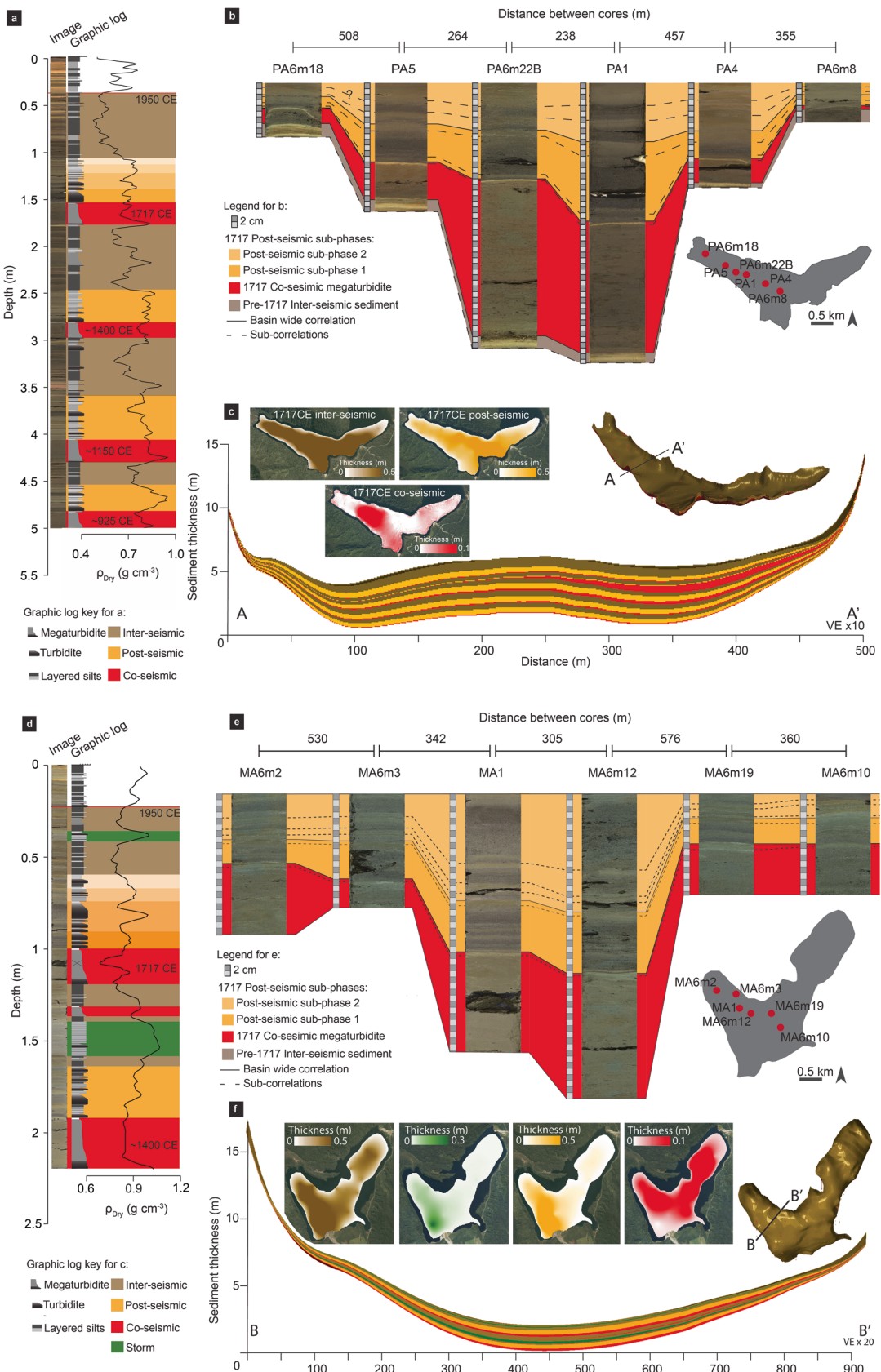

**Fig. 2 | Core sedimentology, inter-core correlations and volume models for lakes Paringa and Mapourika. a**, **d** Images, visual logs and dry density for the master core from the depocenter of each lake. **b**, **e** Examples of the high-resolution correlations that can be traced across the entire lake basins. Lower panels show volumetric models for lakes Paringa (**c**) and Mapourika (**f**). Accumulation over four Alpine Fault earthquake cycles is reconstructed in the Windbag Basin of Lake Paringa, while only two cycles are reconstructed in Lake Mapourika. Insets in b and c show the spatial distribution and thickness of co- (red), post- (yellow), inter-seismic (brown) and storm-induced landslide (green) accumulation over the earthquake cycle between the 1717 CE earthquake and 1950.

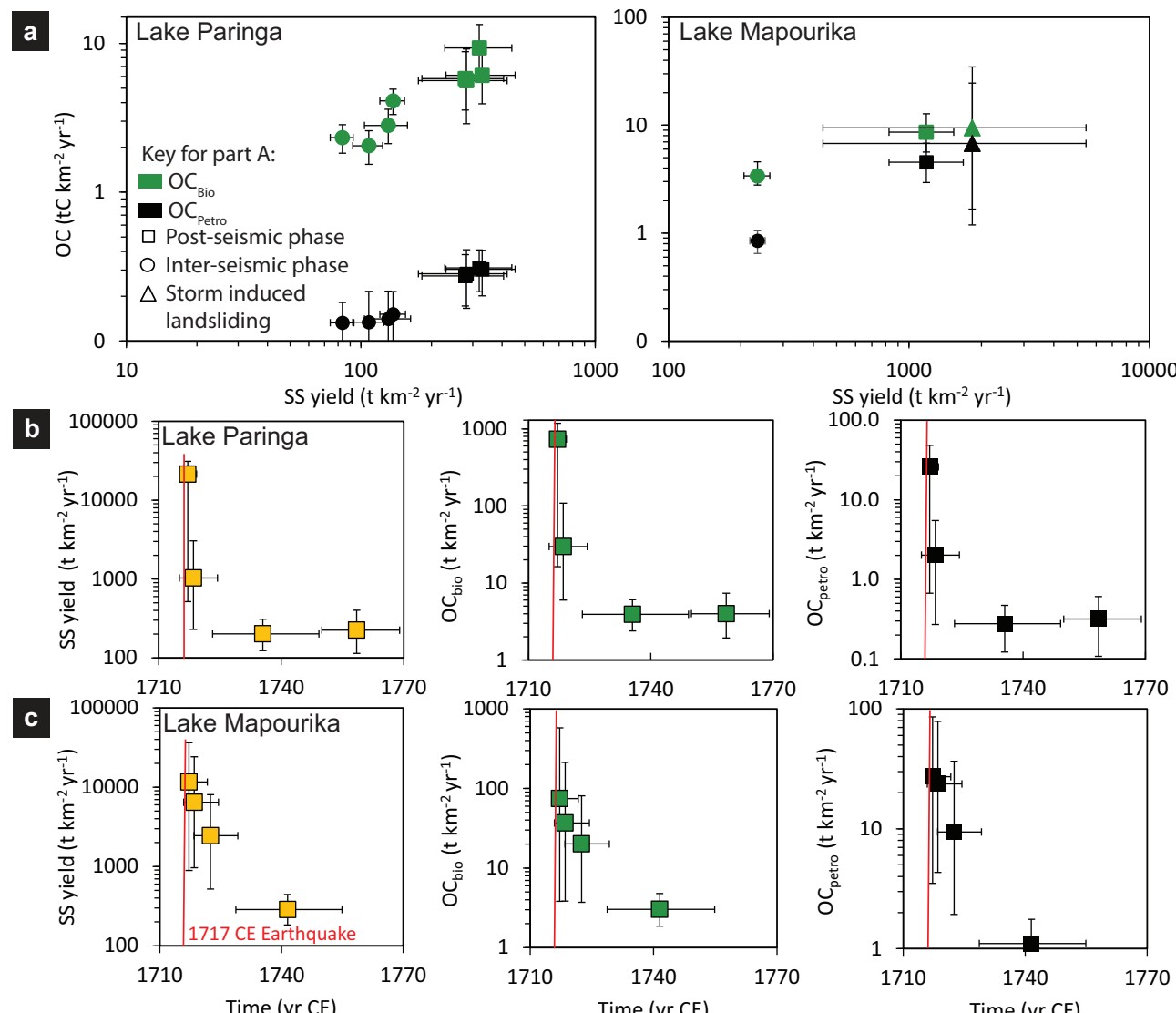

**Fig. 3 | Relationship between suspended sediment (SS), biospheric organic carbon, (OC_bio) and petrogenic organic (OC_petro) yields between post-seismic, inter-seismic and storm-driven landsliding phases.** a Scaling between SS, OC_bio and OC_petro yields for post-seismic (squares), inter-seismic (circles) and storm-driven landsliding (triangles) phases in lakes Paringa and Mapourika. Lower panels show the temporal evolution of SS, OC_bio and OC_petro yields following the 1717 CE earthquake for lakes Paringa (**b**) and Mapourika (**c**). Error bars show 95% Highest Probability Density Function ranges.

$\chi^2 = 24.7$, df = 3, $p < 0.01$; OC_bio yields: $\chi^2 = 12.4$, df = 3, $p < 0.01$). The lowest value for inter-seismic SS yields occurred during the 1400 CE earthquake cycle, while OC_bio yields were lowest during the 1150 CE cycle. Maximum inter-seismic SS and OC_bio yields occurred during the earthquake cycle between the 1717 CE earthquake and 1950.

The average post-seismic SS yields into Lake Mapourika were three times higher than those from Lake Paringa (Fig. 3a). Again, post-seismic sediment yields were significantly higher than the average inter-seismic yields ($\chi^2 = 27.2$, df = 1, $p < 0.01$). There were also two non-earthquake related increases in SS yields, one within each seismic cycle, most likely caused by storm-induced landsliding[24]. These occurred in the late 1800s and mid-1500s and had SS yields that were close to an order of magnitude higher than inter-seismic values. OC_bio yield was only quantified for the earthquake cycle between the 1717 CE earthquake and 1950. For that cycle the post-seismic yield was significantly higher than the inter-seismic yield ($\chi^2 = 8.4$, df = 1, $p < 0.01$), but not substantially different from those recorded in the equivalent phase from Lake Paringa. OC_petro yields were substantially lower than OC_bio yields in both lakes Paringa and Mapourika but recorded similar contrasts between post-seismic and inter-seismic periods.

The precision of the chronology was highest immediately after the 1717 CE earthquake in both lakes, which allowed the most precise yield estimates. Median SS yield peaked in the first year after the earthquake at 21,400 t km$^{-2}$ yr$^{-1}$ (516–31,000 t km$^{-2}$ yr$^{-1}$, henceforth all uncertainties are 95% Highest Probability Density Function (HPDF) ranges unless otherwise stated) in Lake Paringa and 11,600 t km$^{-2}$ yr$^{-1}$ (893–36,400 t km$^{-2}$ yr$^{-1}$) in Lake Mapourika (Fig. 3b, c). The SS yields then decrease to values that were within the 95% HPDF range of the inter-seismic yields in 4 yrs (0.4–10 yrs) in Lake Paringa and 8 yrs (5–16 yrs) in Lake Mapourika. Both OC_bio and OC_petro yields followed similar trends to SS yields in both lakes.

Our reconstructed yields could be biased by incomplete retrieval of the sequence in some cores, erosion of previously deposited material by mass transport processes or heterogeneity in carbon and sediment mass not captured by our sampling approach. Despite these potential sources of error our reconstructed yields agree well with independent denudation rate estimates from cosmogenic nuclides and global scaling between OC_bio, OC_petro and SS yields (Methods, Supplementary Fig. 9). This demonstrates that lake stratigraphic records can preserve quantitative records of yields

from mountain catchments that bridge the gap between instrumental and millennial timescales.

## Implications for extrapolating short-term yields over millennial timescales

Our reconstructions of sediment and carbon yields over decades to a millennium can be used to assess how these yields deviate from the long-term average, defined here as the average yield over the record duration. This informs on the suitability of extrapolating short-term yield estimates from river gauging data to timescales relevant to the geological carbon cycle ($>10^3$ year timeframes). For Lake Paringa the average SS yield over the record duration was 151 t km$^{-2}$ yr$^{-1}$ (138–163 t km$^{-2}$ yr$^{-1}$), while the OC$_{bio}$ yield over the same timeframe was 3.6 t km$^{-2}$ yr$^{-1}$ (3.1–4.2 t km$^{-2}$ yr$^{-1}$). For comparison, the lowest inter-seismic SS and OC$_{bio}$ yields were 0.6 times these long-term averages. Conversely, over the decades where the lake was responding to earthquake-induced landsliding the SS and OC$_{bio}$ yields were up to 2.2 and 1.7 times the long-term yield, respectively. The maximum post-seismic SS and OC$_{bio}$ yields were 142 and 203 times the long-term yield, respectively.

In Lake Mapourika the average SS yield was 434 t km$^{-2}$ yr$^{-1}$ (386–482 t km$^{-2}$ yr$^{-1}$) over the record duration, and the OC$_{bio}$ yield 4.5 t km$^{-2}$ yr$^{-1}$ (3.4–5.6 t km$^{-2}$ yr$^{-1}$) over the last 236 years (233–238 years). Here too, the lowest inter-seismic SS and OC$_{bio}$ yields were a factor of 0.5 and 0.7 times the long-term yields, respectively; while they were up to 4.7 and 2.1 times the long-term yields when responding to earthquake or storm-induced landsliding. The maximum post-seismic SS and OC$_{bio}$ yields were a factor of 27 and 16 times the average long-term yields, respectively. Our reconstructed yields clearly show that yields estimated over short timescales can substantially under- or overestimate the long-term sediment and carbon yields. We therefore recommend caution when extrapolating sediment and carbon yields calculated over years to a decade from river gauging data to longer timescales.

Our approach to reconstructing sediment and carbon yields from lake sediments could be deployed to determine how extreme events influence these yields over timeframes relevant to the carbon cycle in a broader suite of physiographic settings. Lakes have previously been used to quantify changes in sediment yields over post-glacial timescales[27,28] or to anthropogenic land-use change over centuries[22]. However, our study shows that quantitative yield estimates can be generated for the landscape response to discrete extreme events over decadal to millennial timeframes. Lakes preserve diverse extreme events, including earthquakes[29,30], volcanic eruptions[31,32], fires[33,34], storms[32,35,36], and glacial outburst floods, as well as more secular changes in climate and land use change, providing rich avenues for better understanding how these phenomena modulate yields over annual to millennial timescales. Our approach will be most applicable where sedimentary evidence of these extreme events occurs in lakes with simple basin morphology, facilitating more accurate volumetric reconstructions, and the potential to develop high-resolution chronology (e.g. varved lake sediments[37]).

## Extreme event forcing of SS, OC$_{bio}$ and OC$_{petro}$ yields

Previous work has assessed the role of earthquake-triggered landsliding in the carbon cycle for single events in the modern day[18,19] or by relying on changes in accumulation rate in a single core from one lake[38]. Our sediment and OC yields allow us to now quantify how these yields have varied over a millennium, across multiple events. In both lakes Mapourika and Paringa, earthquake and storm-driven landsliding increased yields by up two orders of magnitude over years to decades (Fig. 3). In total, these extreme events caused 42% of SS, 39% of OC$_{bio}$ and 41% of OC$_{petro}$ export to the lake in just 20% of the record duration in Lake Paringa. The same proportions in Lake Mapourika were 54% of SS, 38% of OC$_{bio}$, and 57% of OC$_{petro}$. The similarity between the two catchments demonstrates the importance of pulsed disturbance in driving temporal variation in sediment and carbon yields from the range front of the Southern Alps over century to thousand-year timescales.

These transient increases may have important implications for extrapolating sediment and OC yields derived from short-term instrumental records over longer timeframes if they are representative of mountain catchments more generally[8,39,40]. The transient increase in SS fluxes from extreme event-driven landsliding depends on the ratio of catchment area influenced by the disturbance to the total catchment area[41]. Large earthquakes and high magnitude storms can generate high landslide densities over thousands of square kilometers[42,43]. Hence, we speculate that extreme event-driven landsliding may play an important role in driving transient increases in SS, OC$_{bio}$ and OC$_{petro}$ yields from catchments with areas up to ~$10^3$ km$^2$ that have similar physiography to those studied here. Catchments with areas in the order of $10^2$-$10^3$ km$^2$ dominate the western flank of the Southern Alps and, more generally, the high-standing islands of Oceania[44], where large earthquakes and storms are important drivers of erosion[11,13,14], and short transport pathways transfer highly transient discharges directly to depocenters with little intermediate storage[45]. These small, high-standing, and forested catchments have OC$_{bio}$ yields that are, on average, four times higher than yields from other physiographic settings[2], and have been estimated to produce one-third of organic carbon exports from land to the global ocean[44]. Therefore, our findings, though based on only two catchments, may have broader implications. They suggest SS and OC$_{bio}$ yields determined using river gauging data over years to a decade may underestimate long-term yields from similar catchments in Oceania, which combined are a significant source of OC$_{bio}$ export to the global ocean.

Our findings may be less relevant for larger continental-scale catchments. In these settings, where catchment areas exceed ~$10^4$ km$^2$, the spatial extent of event-driven landsliding is often less than the total catchment area (e.g. ref. 18). If the ratio of disturbed catchment area to the total catchment area controls the magnitude of transient increases in yield[41], then transient increases in OC$_{bio}$ export caused by extreme-event driven landsliding may not be as pronounced. These systems also have longer sediment and carbon residence times that may result in higher rates of OC$_{bio}$ oxidation as it passes through long floodplains before eventual burial in the ocean[10,46]. For example, the 32,000 km$^2$ Narayani R. catchment draining the Nepalese Himalaya showed a negligible increase in SS or OC$_{bio}$ yields following the 2016 $M_w$7.8 Gorka earthquake that triggered extensive landsliding[18]. There, co-seismic landslides impacted the eastern part of the catchment and represented only a modest increase over annual monsoon-driven rates of landsliding[47]. Determining the suite of physiographic settings where extreme events drive substantial temporal variation in yields requires future research that applies our reconstruction approach in more diverse settings.

To provide a first-order estimate of the global importance of earthquake-induced landslides (EQIL) as a control on OC$_{bio}$ erosion, we combine an EQIL database[48] with the global mean biomass and soil stocks (Methods). A global database of rainfall-induced landsliding does not exist, precluding an equivalent analysis of that trigger[49]. The 38 mapped events of EQIL have disturbed a total area of 5748 km$^2$ globally between 1976 and 2019, at a rate of 133 km$^2$ yr$^{-1}$ (ref. 48). A biome-based classification of vegetation and soil carbon stocks is used to estimate the OC$_{bio}$ mobilized by these landslides[50]. We estimate a total of 103 MtC (43 MtC and 60 MtC mobilized from vegetation and soil, respectively) by these events. This represents an average OC$_{bio}$ yield by EQIL of 2.4 MtC.yr$^{-1}$. The uncertainties on this estimate are difficult to quantify because events may be missing from the EQIL database[48], while temporal variations in EQIL rate are not likely to be captured by the ~50-year timeframe. Nevertheless, this flux is similar in magnitude to the annual OC$_{bio}$ flux from the Ganges – Brahmaputra, one of the largest sources of OC$_{bio}$ to the global Ocean[8]. EQIL may thus be a globally significant driver of OC$_{bio}$ erosion relevant to the sedimentary carbon cycle if this OC$_{bio}$ is effectively exported from hillslopes to ocean depocentres. Our analysis does not constrain how much earthquake-mobilised OC$_{bio}$ is oxidised as it traverses the transport pathway from hillslopes to the ocean[17]. Modelling approaches that capture sediment transport and transient storage[17] have suggested that even with rapid organic matter degradation, the majority of earthquake mobilized OC$_{bio}$ can be exported from mountain catchments, yet the detailed fate of widespread-landslide mobilized sediment and organic carbon through downstream fluvial systems remains a major uncertainty[10].

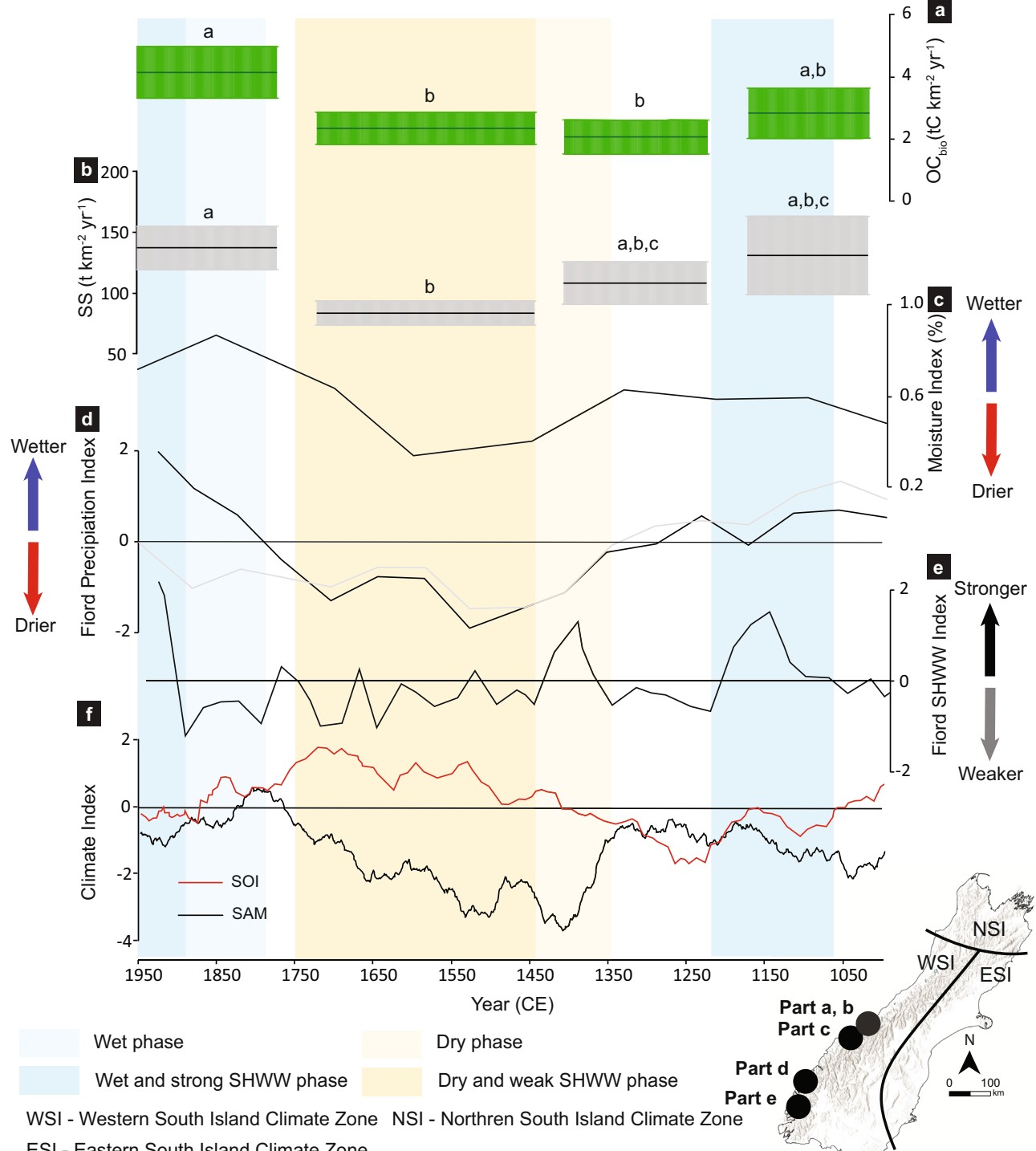

**Fig. 4 | Inter-seismic biospheric organic carbon (OC_bio) yields compared to reconstructed hydroclimate, Southern Hemisphere Westerly Wind (SHWW) intensity, and related climate indices. a** Inter-seismic $OC_{bio}$ yields, and (**b**) sediment yield (SS) from Lake Paringa (solid lines indicate medians; shaded regions represent 95% Highest Probability Density Function ranges; phases sharing the same letter are not significantly different based on Bonferroni-corrected pairwise chi-squared tests). **c** Reconstructed precipitation indices derived from pollen records[25], and **d** fiord carbon stable isotope ratios from the Western South Island Climate Zone (WSI)[51]. **e** Reconstructed SHWW intensity over southern New Zealand[52], a primary driver of positive precipitation anomalies in the WSI during the instrumental period[92,93]. **f** Variations in SHWW intensity driven by interactions between the Southern Annular Mode (SAM; black line)[53] and the Southern Oscillation Index (SOI; red line)[54], which together enhance SHWW intensity over southwest New Zealand when both SAM and SOI are in phase and negative[55].

## Climate drivers of inter-seismic SS and OC_bio yields

Hydroclimatic variability has been invoked as an important driver of sediment and carbon yields in mountain rivers[9] based on space for time substitution experiments from global river yield datasets. Here, the thousand-year timeframe over which the Lake Paringa record covers provides an opportunity to test this hypothesis. Indeed, we find there is a significant difference in SS and $OC_{bio}$ yields between at least one of the individual inter-seismic phases of the Lake Paringa record i.e. the time periods when the Lake Paringa catchment is not responding to enhanced SS and $OC_{bio}$ yields from earthquake-induced landsliding (Fig. 4). In Lake Mapourika, the shorter

length of the $OC_{bio}$ yield record (a single inter-seismic period) precludes any assessment of temporal variability.

Paleoclimate reconstructions from within the same regional climate zone as Lake Paringa show rainfall[25,51] and its main driver Southern Hemisphere Westerly Wind (SHWW) intensity[52] co-vary with $OC_{bio}$ and SS yields (Fig. 4). While the low number of inter-seismic yield estimates and variable temporal resolution of the paleoclimate data preclude formal time series analysis, first-order qualitative trends are clear. $OC_{bio}$ yields were lowest (between $2.1 \pm 0.5$ and $2.3 \pm 0.5$ tC km$^{-2}$ yr$^{-1}$) during the drier period of the record when SHWW intensity over the southern South Island was low. Conversely, the highest inter-seismic $OC_{bio}$ yields (between $4.1 \pm 0.8$ and $2.8 \pm 0.8$ tC km$^{-2}$ yr$^{-1}$) were found during wetter periods when SHWW intensity was higher. Precipitation and SHWW location and intensity are influenced by interaction between the Southern Annular Mode (SAM)[53] and Southern Oscillation Index (SOI)[54] that cause higher precipitation and SHWW intensity over southwest New Zealand when both SAM and SOI are in phase and negative[55] (Fig. 4). Time intervals where these climate indices were in phase and negative correspond to periods when reconstructed sediment and $OC_{bio}$ yields are higher, providing further support for the link with climatic change. This suggests that precipitation moderates $OC_{bio}$ export over century to millennial timescales during time periods when the landscape is not responding to large magnitude earthquakes and storms. As precipitation is strongly correlated with runoff in the high precipitation regime of the Southern Alp's western flank[56], our findings also support hypotheses from space for time substitution experiments that use $OC_{bio}$ yield from river gauging and spatial gradients in climate to show runoff controls $OC_{bio}$ export[9].

To further explore the inter-seismic data from Lake Paringa, we use a runoff-driven erosion model of $OC_{bio}$, which has previously been used to describe the climate sensitivity of $OC_{bio}$ export in mountain catchments[9]. Informed by a shear-stress erosion model and fit to empirical data from mountain river catchments globally, river particulate $OC_{bio}$ concentrations ($[POC_{bio}]$, mg L$^{-1}$) are described by a function of instantaneous runoff, $R$ (mm day$^{-1}$):

$$[POC_{bio}] = \alpha R^{\gamma}$$

where $\alpha$ (units in of mg L$^{-1}$ mm$^{-1\gamma}$ day$^{\gamma}$) is a pre-factor that is proposed to be related to the slope angle of the catchment[9], and the exponent $\gamma = 1.37 \pm 0.17$ (95% CI)[9]. To provide a first order estimate of fluxes and the controls, we assume $\alpha = 0.052$, which reflects the Lake Paringa catchment with some areas of lower slope compared to trunk valley catchments[9]. We use a mean annual runoff of $3848 \pm 440$ mm yr$^{-1}$ (95% CI) calculated from the Paringa catchment rain gauge[57], and corrected for the trend of increasing rainfall between 1960 and 2022 of $154 \pm 114$ mm (95% CI) per decade in the Western South Island Climate Zone[58]. The model returns an $OC_{bio}$ concentration of $1.3 \pm 0.6$ (95% CI) mgC l$^{-1}$, which equates to an annual load of $201 \pm 87$ (95% CI) tC yr$^{-1}$ when multiplied by the annual flow (l yr$^{-1}$), and an $OC_{bio}$ yield of $5.0 \pm 2.2$ (95% CI) tC km$^{-2}$ yr$^{-1}$ when normalised by catchment area. The modelled yield agrees within uncertainty with the lake-derived estimate of $4.1 \pm 0.8$ tC km$^{-2}$ yr$^{-1}$ (Fig. 4a) for the most recent inter-seismic period (1770–1950). The slightly lower central tendency of the reconstructed yield can be explained by material not trapped by the lake, which has a trapping efficiency of 92% (Methods).

Building on the observed relationship between reconstructed $OC_{bio}$ yields and hydroclimate, combined with the agreement between reconstructed and modelled yields, we use the runoff-driven model to quantify the change in $OC_{bio}$ export predicted for future climate change. For a mid-range scenario (i.e. RCP6.0), streamflow in the western Southern Alps is projected to increase by 10–20% from the 1986–2005 average to 2080–2099 (ref. 59). Based on the parameterized runoff-erosion model used here, $OC_{bio}$ export could increase by $39 \pm 30\%$ (95% CI) over this timeframe (from $5.0 \pm 2.2$ tC km$^{-2}$ yr$^{-1}$ to $7.0 \pm 4.5$ tC km$^{-2}$ yr$^{-1}$). Our analysis is likely conservative because it doesn't account for the predicted increase in severity of the largest flow events[59], which will amplify $OC_{bio}$ erosion because it scales non-linearly with runoff[9,16]. These events are also more likely to cause widespread land-sliding, which we have shown results in dramatic increases in $OC_{bio}$ yields.

## Implications for the carbon cycle

The erosion of $OC_{bio}$ from the terrestrial biosphere and transfer into sedimentary storage acts as a net $CO_2$ sink and $O_2$ source over geological time[8,9]. Nevertheless, it still remains unclear how orogenesis, erosion and weathering contribute to the carbon cycle, and the feedbacks which link surface processes to climate change[2], which is usually viewed through the lens of the silicate weathering feedback (e.g. ref. 60). In our reconstructed timeseries spanning the last millennium we see an important control on $OC_{bio}$ export by active tectonics. Co-seismic landsliding and subsequent sediment and $OC_{bio}$ export is responsible for ~40% of the $OC_{bio}$ and SS transfer in these steep forested catchments. However, during the inter-seismic periods that dominate total mass export from these catchments (~60% of the long-term flux in Lake Paringa), the fluxes are mediated by hydroclimatic variability. Wetter periods have higher $OC_{bio}$ yields, and the most recent climate change from the Little Ice Age (~1450 CE–1850 CE) coincided with a doubling of $OC_{bio}$ fluxes into the lake.

Our reconstructions support the hypothesis based on global scale space for time substitution experiments on modern observations that erosion of $OC_{bio}$ could act as a negative feedback in the Earth System[9]. Wetter climates associated with warmth can enhance erosion of $OC_{bio}$ from terrestrial ecosystems, promote its transfer into long-term sedimentary storage, and drawdown $CO_2$ from the atmosphere (Fig. 4). The lake timeseries is consistent with an estimated ratio between changing runoff and $CO_2$ drawdown fluxes from $OC_{bio}$ export and burial of ~1:1.6 (ref. 9). By comparison, a ratio of <1:0.6 (ref. 61) is thought to be typical of silicate weathering which is the other key negative feedback on geological timescales[60,62]. If temperature and hydroclimate are coupled at the global scale over geological timescales[63], then $OC_{bio}$ export and burial represents a sensitive negative feedback that is not currently accounted for in Earth System models that seek to quantify the carbon cycle[64]. Understanding the operation of the geological carbon cycle once terrestrial ecosystems established in the Phanerozoic and erosion provided an additional negative feedback will require concerted research effort.

## Methods
### Study site

The Southern Alps are formed by oblique convergence between the Australian and Pacific plates at a rate of 39.7 mm yr$^{-1}$ on a bearing of 245° (ref. 65). Up to 80% of the plate motion is accommodated on the range-bounding Alpine Fault[66], which produces major and great earthquakes ($M_w$ >7) with a return period of $251 \pm 48$ (1$\sigma$) years[23]. The western flank of the Southern Alps is dominated by steep slopes developed in tectonically fractured metasedimentary bedrock that support high rates of landsliding[67–69]. Hillslopes are covered in largely undisturbed temperate rainforest below a treeline at ~1000 m. High rates of net ecosystem productivity of $94 \pm 11$ (2SE) tC km$^{-2}$ yr$^{-1}$ support carbon stocks of $17,500 \pm 5500$ (2SE) tC km$^{-2}$ in above-ground biomass and $18,000 \pm 9000$ (2SE) tC km$^{-2}$ in soils[70]. Moisture derives predominantly from the Tasman Sea[71] and is transported by north westerly flows that result in orographic precipitation of between 5 m yr$^{-1}$ and 12 m yr$^{-1}$ (ref. 72) in the present day. Paleoclimate reconstructions show late Holocene variation in hydroclimate with a pronounced precipitation minimum between 1750 CE and 1350 CE[25,51]. The climate and tectonic setting produce erosion rates of up to 10 mm yr$^{-1}$ (refs. 73,74). Lakes Paringa and Mapourika are located ~3–5 km west of the Alpine Fault, have range front catchments with areas of ~60 km$^2$ (Fig. 1). Erosion in these catchments occurs predominantly (~90%) on slopes above 15°[26], so contributions from alluvial fans and floodplains is minimal.

### Sediment coring and analysis

The sediment coring approach aimed to provide unprecedented 4D resolution on the sediment fill in these lakes. A Mackereth corer was used to retrieve sediment cores up to 6 m long from 28 sites in Lake Paringa and 22

in Lake Mapourika[75], providing a spatial coverage of approximately one core per ~500 m (Fig. 1 and Supplementary Fig. 1). Sediment cores were digitally imaged using a GEOTEK linescan camera, logged visually and X-ray computed tomography (CT) scanned using a GE BrightSpeed medical CT scanner set to 120 kV, 250 mA, pitch of 0.625 mm and a 100 cm² window. Stratigraphic correlations between cores were achieved by mapping litho-facies stacking patterns that are easily identified in digital imagery and CT tomography and can be traced laterally throughout the lake basins due to the close core spacing[24] (Fig. 2). These correlations were mapped in Corelyzer 2.1.2[76] and validated using radiocarbon dates to produce basin-wide fence diagrams of contacts between co-seismic, post-seismic, and inter-seismic sedimentary units (that we subsequently term phases) to inform volumetric and density modelling of the sedimentary fill (Fig. 2a, c; Supplementary Figs. 2 and 3). Higher resolution intra-phase correlations were also achieved within the 1717 CE post-seismic phase and during inter-seismic phases that contained periods of increased sediment and OC yields from storm-induced landsliding.

In Lake Paringa we focused the coring and sedimentary fill volume modelling on the Windbag Basin. The major fluvial tributaries of the lake flow into this basin and it is where the majority of terrestrial sediment accumulates in the lake[11,23] (Supplementary Fig. 1). Cores from the sill that separates the Windbag Basin from the Hall Basin show minimal accumulation, which suggests that the amount of material that enters the Hall Basin from the Windbag Basin is negligible. This observation supports our decision to model volumes and yields for the Windbag basin alone, though we acknowledge that our yield estimates from Lake Paringa are conservative.

### Lake core chronology

Chronology was generated for master cores located in the depocentre of each lake (PA1 and MA1) using [137]Cs, radiocarbon dating and earthquake age estimates from Howarth et al.[23]. [137]Cs was measured using gamma spectrometry at the Institute of Environmental Science and Research, Christchurch, New Zealand with a high-purity germanium well detector. Activities are reported in Bq.kg[−1] as mean values with 95% confidence limits (Supplementary Data 4). Radiocarbon dates on terrestrial macrofossils (23 for Paringa and 15 for Mapourika) were treated and measured by Accelerator Mass Spectrometry (AMS) according to the methods of Baisden et al.[77] at Rafter Radiocarbon Laboratory and Fink et al.[78] at ANSTO AMS facility and are reported in Howarth et al.[23]. The radiocarbon dates were calibrated with the ShCal20 calibration curve[79]. Independent age estimates for each earthquake were derived from Wells et al.[80] for the 1717 CE event and Howarth et al.[23] for the ~1400 CE, ~1150 CE and ~925 CE earthquakes, and were used as additional temporal constraints in the age-depth models. All chronological information was integrated with core depth using the P_Sequence prior model with an event thickness constant k derived empirically (cf. Ramsey[81]; k = 2 for Paringa and k = 3 for Mapourika) in OxCal 4.4 to produce an age depth model for each lake[81] (Supplementary Fig. 6). Depth intervals associated with rapidly deposited layers (RDL) and voids generated by gas were removed from the depth sequence to produce RDL-corrected depth that was used for age modelling purposes because the formation of RDL is near instantaneous and they do not represent the passage of time. The age models were used to derive the duration and related uncertainty of post-seismic, inter-seismic and storm-induced landslide phases and sub-phases that were then used in the production of SS, OC_bio and OC_petro yields.

### Lake sediment volume modelling and sediment yield estimates

The volumes of co-, post- and inter-seismic phases as well as 1717 CE post-seismic and storm-induced landslide sub-phases were modelled using lake bathymetry[82,83] and down core logs in the 3D geological mapping software Leapfrog™ using radial basis function interpolation in the Offset Surface tool[84] (Supplementary Figs. 4 and 5; Supplementary Movies 1 and 2). The boundaries of the model were set using bathymetric slopes of 20°, above which geophysical data revealed that minimal sediment accumulates. 3D

models of sediment bulk density variation across the sedimentary fill were also generated in Leapfrog using the downcore sediment bulk density derived from the CT tomography[85].

The wet mass ($\Upsilon_w$, kg) of all sedimentary phases and sub-phases was calculated using:

$$\Upsilon_w = \sum_{i=1}^{n} V_i \rho_i \qquad (1)$$

where $V_i$ (m³) is the volume of a given voxel (5 m³) in the Leapfrog volume model for a phase and $\rho_i$ is the bulk (kg m⁻³) density of that voxel derived from the Leapfrog bulk density model.

The dry mass ($\Upsilon_d$, kg) of all sedimentary phases and sub-phases was then calculated using:

$$\Upsilon_d = \Upsilon_w \left(1 - \frac{W}{100}\right) \qquad (2)$$

and $W$ is the average weight percent water content of the phase derived using mass loss on drying at 105 °C from samples from the master cores from each lake (Supplementary Data 1). Spatial variability in water content was quantified for each phase over the 1717 CE seismic cycle in each lake and showed that the standard errors of the mean (SE) $W$ for each phase were always less than 2.5%. Consequently, a conservative uncertainty for the $W$ of each phase of ±5% (2SE) was adopted and propagated in quadrature through $\Upsilon_d$ calculations. We make no correction for trapping efficiency because it is estimated to be 92% based on the method of Gill[86] that uses the ratio of lake capacity to mean annual inflow.

Suspended sediment mass per unit catchment area and per unit time (SS, expressed as t km⁻² yr⁻¹) were generated using:

$$SS = \Upsilon_d / A / t \qquad (3)$$

where $\Upsilon_d$ is the dry mass of the phase or sub-phase, $A$ is the catchment area with hillslope angle above 15° (~40 km² for Lake Paringa and ~20 km² for Lake Mapourika), and $t$ is the phase duration derived from the chronology (Supplementary Data 2). Chronological and $\Upsilon_d$ uncertainties were propagated through the calculations using a Monte Carlo approach in OxCal 4.4 and reported as median values with 95% highest probability density function (HPDF) ranges unless otherwise stated.

### Lake sediment geochemical analysis and OC yield estimates

The average OC content (weight %) and carbon stable isotope ratios of each depositional phase or sub-phase were quantified using samples from core PA1 in Lake Paringa and core MA6m1 from Lake Mapourika that were processed and measured by EA-IRMS according to the methods of Frith et al.[38]. For Lake Mapourika, we focus on the post 1717 CE section in MA6m1 (Supplementary Data 3). In addition, we have measured the radiocarbon activity of bulk organic carbon in MA6m1 ($n = 10$) and catchment soils ($n = 6$) using methods described by Wang et al.[26], reported as the fraction modern, $F^{14}C$ (Supplementary Data 3).

Total OC ($OC_{total}$, tC) of each phase or subphase was estimated using Eq. (3)

$$OC_{total} = \Upsilon_d \left(\frac{OC\%}{100}\right) \qquad (4)$$

where $\Upsilon_d$ (t) is the dry sediment mass and OC% is the weighted average organic carbon weight percent of the phase. Uncertainties in $\Upsilon_d$ and OC% (2SE) were propagated in quadrature through the calculation of $OC_{total}$. $OC_{total}$ from each phase was partitioned into $OC_{bio}$ and $OC_{petro}$ fractions. For Lake Paringa, the fraction of rock-derived organic carbon ($F_{petro}$) was quantified using measured total organic carbon (TOC) of the lake sediment samples and a binary mixing model with a bedrock end-member (0.13 ± 0.1% (2SE)) as discussed in previous work[38,87]. This was appropriate

because of the high OC% in the lake sediments (mean = 2.64 ± 0.55% (2SE)), which preclude significant inputs of $OC_{petro}$.

In contrast, the average OC% in Lake Mapourika was lower (mean % OC = 1.38 ± 0.18% (2SE)), meaning $OC_{petro}$ may contribute more to the carbon mass in the core. As such, the $F_{petro}$ for Lake Mapourika was constrained using a binary mixing model based on $^{14}C$ fraction modern measurements ($F^{14}C$, $n = 10$; Supplementary Fig. 7), where the $OC_{petro}$ end member is 0 and the $OC_{bio}$ end member is derived from measurements of soil O horizons[26]. In this subset of the samples, we find that $F_{petro}$ correlates with $\delta^{13}C$ in the lake core, reflecting the fact that the $OC_{petro}$ end member has a high $\delta^{13}C$ value, distinct from the vegetation[88]. As such, we use a linear regression between $F_{petro}$ and $\delta^{13}C$ ($r^2 = 0.84$) to apply to the wider sample set for which we have $\delta^{13}C$ values but not $F^{14}C$ (Supplementary Fig. 8). Uncertainty of two times the standard error of the predicted value was propagated in quadrature through the calculation of $F_{petro}$ for Lake Mapourika. For both lakes uncertainty derived from $F_{petro}$ estimates (2SE) was propagated in quadrature through the calculation of $OC_{bio}$ and $OC_{petro}$ masses. We assume negligible lake-derived (autochthonous) $OC_{bio}$ because biomarker and bulk sediment stable isotope analysis indicate only minimal $OC_{bio}$ from aquatic biomass[38].

$OC_{bio}$ and $OC_{petro}$ yields per unit catchment area and per unit time were calculated using equations analogous to Eq. (2) (Supplementary Data 2). Uncertainties for $OC_{bio}$ and $OC_{petro}$ masses and the chronology were propagated through calculations using Monte Carlo sampling and reported as median values with 95% highest probability density function (HPDF) ranges unless otherwise stated. Statistical differences between distributions of SS, $OC_{bio}$, and $OC_{petro}$ yields were assessed using chi-squared ($\chi^2$) tests of depositional phase means and standard deviations under the assumption of approximate normality[76]. A global chi-squared test evaluated agreement among all phases ($\alpha = 0.05$), while Bonferroni-corrected pairwise tests identified specific differences ($\alpha = 0.0083$).

### The accuracy of reconstructed sediment and organic carbon yields

In lieu of overlapping instrumental and reconstructed timeseries the accuracy of our reconstructed yields was assessed using comparisons to independent denudation rate estimates from cosmogenic nuclides and global scaling between SS, $OC_{bio}$, and $OC_{petro}$ yields. $^{10}Be$-derived denudation yields from two range front catchments near Lake Mapourika with similar physiographic characteristics (morphometry, rock type, vegetation and climate) as Potters Creek provide a mean yield of $4870 ± 1340$ t km$^{-2}$ yr$^{-1}$ (2SE) averaged over $485 ± 182$ years (2SE)[89]. This estimate is an order of magnitude greater than the reconstructed SS yields over the same timeframe from Lake Mapourika, which is expected because the coarse sediment and dissolved loads are not captured in our lake-based reconstruction. If we take the reconstructed SS yield from Lake Mapourika, we can estimate the missing coarse sediment load component assuming that: 1) landsliding drives erosion and sediment production in the Southern Alps[68]; 2) $12 ± 6\%$ (2SE) of landslide grainsize distributions are <2 mm based on landslide grainsize measurements from a similar physiographic setting (New Zealand and Taiwan)[90,91], and 3) chemical denudation is an order of magnitude less than physical denudation and hence a small component of total denudation in these systems[89]. The total sediment yield for the Lake Mapourika catchment in this case would be $3630 ± 1880$ t km$^{-2}$ yr$^{-1}$ (2SE), which is similar to the $^{10}Be$ denudation rates. Unfortunately, there is no comparable data for validating the Lake Paringa SS yields, though the lower yields compared to Lake Mapourika are comparable to the reduction in landslide-derived erosion rates in the Southern Alps between these loctions[68], and consistent with a decrease in mean annual precipitation[72] moving south west.

The lake-derived estimates of SS, $OC_{bio}$ and $OC_{petro}$ yield are also consistent with relationships derived from modern-day fluxes in global rivers[8,9] (Supplementary Fig. 9). Our estimates of $OC_{bio}$ yields for a given SS yield are within the range of global values, though they sit above the regression line[8]. This is expected given the high organic carbon stocks of soils in the western Southern Alps[70], which results in high $OC_{bio}$ yields when

mobilized by landsliding[88]. The reconstructed $OC_{petro}$ yields for a given SS yield fall closer to the global regression line than the $OC_{bio}$ yields, probably because rock OC% of Southern Alps bedrock is close to the average for most major rock types globally[5].

### Global earthquake-induced landslide mobilized $OC_{bio}$ estimate

We use the EQIL ("earthquake induced landslide") database of Seal et al.[48] and filter it for events where landslide area has been mapped. Since the satellite era, 38 earthquakes have been studied and landslide maps produced between 1976 and 2019. Many more EQIL events have metrics such as the total impacted area, and number of landslides, but we have focused only on those with reported landslide areas. This approach is likely to have missed events over this time interval. We recognize that the time period is short compared to the typical return interval of earthquakes and the full magnitude frequency distribution of events will not have been sampled.

To calculate the eroded mass of organic carbon from soil and vegetation ($OC_{bio}$) we follow the approach of Hilton et al.[70], who use average carbon stocks for soils and vegetation. Because the landslide event locations are linked to the earthquake epicenter in the EQIL database[33] and that individual landslide polygons are not available, we assess the carbon stocks for an event using biome estimates of organic carbon stocks in vegetation and soils from Watson et al.[50]. As such, we report a total mobilized mass and do not focus on individual events which may represent over- or underestimates based on the local soil and vegetation carbon stocks.

## Data availability

The datasets that support the findings of this manuscript, including core to core correlations, and the volumes and density models are available in figshare project 'Long term carbon export from mountain forests driven by hydroclimate and extreme event driven landsliding' (https://figshare.com/projects/Long_term_carbon_export_from_mountain_forests_driven_by_hydroclimate_and_extreme_event_driven_landsliding/125050).

## Code availability

The OxCal 4.4 code used for the age modelling and Monte Carlo analysis of yields is also available in figshare project (https://figshare.com/projects/Long_term_carbon_export_from_mountain_forests_driven_by_hydroclimate_and_extreme_event_driven_landsliding/125050).

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

## Acknowledgements

Natural Environment Research Council Standard Grant to R.G.H., J.D.H. and A.L.D. (NE/P013538/1) funded the sediment coring, organic geochemistry and radiocarbon measurements. J.D.H. acknowledges funding from the Leverhulme Trust, which facilitated a Visiting Professorship at the University of Oxford. Lake Sediment cores were collected under Department of Conservation (DoC) research permit 38491-RES. We thank Delia Strong for assistance in the field.

## Author contributions

J.D.H. and R.G.H. designed the study. S.J.F. and J.D.H. collected the sediment cores. J.D.H., A.M. and S.J.F. generated and interpreted the sedimentological and chronological data. A.M., J.D.H. and A.H. produced the volume and density models. J.W., R.G.H., J.D.H., A.L.D., T.C, and M.G. produced and interpreted the organic geochemistry and [14]C data. J.D.H. and R.G.H. wrote the paper with contributions from all the co-authors.

## Competing interests

The authors declare no competing interests.

**Additional information**

