## [Transparent Peer Review file · Communications Earth & Environment]

Long-term carbon export from mountain forests driven by hydroclimate and extreme event-driven landsliding

Corresponding Author: Dr Jamie Howarth

Version 0:

Decision Letter:

Dear Dr Howarth,

Your manuscript titled "Long term carbon export from mountain forest driven by changing hydroclimate & landsliding" has now been seen by 2 reviewers, and we include their comments at the end of this message. They find your work of interest, but some important points are raised. We are interested in the possibility of publishing your study in Communications Earth & Environment, but would like to consider your responses to these concerns and assess a revised manuscript before we make a final decision on publication.

We therefore invite you to revise and resubmit your manuscript, along with a point-by-point response that takes into account the points raised. Please highlight all changes in the manuscript text file.

We also ask that you address the following editorial thresholds;

* Please explain how the organic carbon transported from landslides to lake sediments can be scaled up to the global carbon cycle, despite the localised nature of the study. Your explanation should highlight the mechanisms and processes that allow for this extrapolation, such as the frequency and distribution of landslides globally, the role of lakes as carbon sinks, and the integration of local data into global carbon models.

* Please include a section that discusses how extreme events, like landslides, can have a smoothing effect on carbon records over time. Include a discussion point on whether lake sediment records from various global locations, if sufficiently long, can provide reliable insights into long-term trends in carbon cycling.

Please submit your point-by-point responses as a separate file, distinct from your cover letter where you can add responses to the Editors' comments that you do not want to be made available to the reviewers. Word files are preferred. We recommend that any figures, tables or graphs that are included in the response to reviewers are also included in the main article or Supplementary Information.

Please use the following link to submit your revised manuscript, point-by-point response to the referees' comments (which should be in a separate document to any cover letter), a tracked-changes version of the manuscript (as a PDF file) and the completed checklist:

Link Redacted

We hope to receive your revised paper within six weeks; please let us know if you aren't able to submit it within this time so that we can discuss how best to proceed. If we don't hear from you, and the revision process takes significantly longer, we

may close your file. In this event, we will still be happy to reconsider your paper at a later date, as long as nothing similar has been accepted for publication at Communications Earth & Environment or published elsewhere in the meantime.

Please do not hesitate to contact us if you have any questions or would like to discuss these revisions further. We look forward to seeing the revised manuscript and thank you for the opportunity to review your work.

Best regards,

Adam Switzer, PhD
Editorial Board Member
Communications Earth & Environment
orcid.org/0000-0002-4352-7852

Carolina Ortiz Guerrero, PhD
Associate Editor
Communications Earth & Environment

EDITORIAL POLICIES AND FORMATTING

Editorial Policy: [Policy requirements](https://www.nature.com/documents/nr-editorial-policy-checklist.pdf) (Download the link to your computer as a PDF.)

- Behavioural and social science
- Ecological, evolutionary & environmental sciences
- Life sciences

<https://www.nature.com/documents/nr-reporting-summary.zip>

Furthermore, please align your manuscript with our format requirements, which are summarized on the following checklist: [Communications Earth & Environment formatting checklist](https://www.nature.com/documents/commsj-phys-style-formatting-checklist-article.pdf)

and also in our style and formatting guide [Communications Earth & Environment formatting guide](https://www.nature.com/documents/commsj-phys-style-formatting-guide-accept.pdf) .

*** DATA: Communications Earth & Environment endorses the principles of the Enabling FAIR data project (<http://www.copdess.org/enabling-fair-data-project/>). We ask authors to make the data that support their conclusions available in permanent, publically accessible data repositories. (Please contact the editor if you are unable to make your data available).

All Communications Earth & Environment manuscripts must include a section titled "Data Availability" at the end of the Methods section or main text (if no Methods). More information on this policy, is available at <http://www.nature.com/authors/policies/data/data-availability-statements-data-citations.pdf>.

If a community resource is unavailable, data can be submitted to generalist repositories such as [figshare](https://figshare.com/) or [Dryad Digital Repository](http://datadryad.org/). Please provide a unique identifier for the data (for example a DOI or a permanent URL) in the data availability statement, if possible. If the repository does not provide identifiers, we encourage authors to supply the search terms that will return the data. For data that have been obtained from publically available sources, please provide a URL and the specific data product name in the data availability statement. Data with a DOI should be further cited in the methods reference section.

Please refer to our data policies at <http://www.nature.com/authors/policies/availability.html>

REVIEWER COMMENTS:

Reviewer #1 (Remarks to the Author):

This is a review report for the manuscript, entitled: Long term carbon export from mountain forests driven by changing hydroclimate and landsliding, by Dr. Howarth et al. This study took many cores in Lake Paringa and Mapourika, New Zealand and reconstructed the SS, OC_{bio}, and OC_{petro} export for millennial timescales. They found that Landsliding triggered by earthquakes or rainstorms plays a dominant role in OC export, which can be an additional CO₂ drawdown, even overwhelms silicate weathering. Moreover, after episodic landsliding, the elevated OC_{bio} export which can persist for a long time (at least 4-10 yr, even longer) is also significant negative feedback. The former argument has been discussed/published/validated for some recent studies. The later argument (prolonged elevated OC_{bio} export) is very innovative.

I have two concerns before suggesting to the next publication stage.

The first one is that the sequence of OC_{bio} from landslides to transport to burial (lake sediments) has been demonstrated by authors. The signal or amount of OC_{bio} in this case may be special due to the short pathway. The longer pathway or travel time enhances OC_{bio} decomposition. With the increase of travel time, the increasing OC_{bio} decomposition may hinder the negative feedback. At global scale, I am curious how to take this into considering global riverine OC_{bio} export [Line: 220-233]

The second one is Fig.4:

As core and dating data showed an extreme earthquake occurred in ~1717CE and buried a lot of OC_{bio} (Fig. 2). I just wonder why this can't be found in Fig. 4(a)? It looks like a long-term average. According to Fig.3, I think you can have more than 10 dots for Fig. 4(a). Even the OC_{bio} exports are different in the two lakes, I think their changes are consistent.

Minor comments:

What do the lines in supplementary Fig. 9 stand for? In the Fig. 9(a), is it a regressive line? If they are regressive line, please provide the r^2 and p-value.

Reviewer #2 (Remarks to the Author):

Review of COMMSENV-24-2345-T: "Long term carbon export from mountain forests driven by changing hydroclimate & landsliding" by J. Howarth and colleagues

Dear Authors,

In this contribution you estimate the flushing of sediment and organic carbon (OC) from small catchments draining the western Southern Alps of New Zealand. You argue that rainstorms and strong earthquakes are the main processes responsible for episodically eroding densely forested hillslopes, and thus delivering large amounts of OC to two foreland lakes that serve as sediment traps and archive such pulses. Drawing on your previous research, in which you identified several cycles of sedimentation following major earthquakes, you offer here mass estimates of material deposition during co-, post-, and interseismic intervals over the past millennium. You argue that strong earthquakes and rainstorm-induced episodes of sediment influx to the lakes have average OC yields that outweigh those during longer periods of relative geomorphic quiescence. You also propose that palaeoclimatic proxies of regional precipitation or wetness help to explain variations in the inferred sediment and OC yields. You conclude by pointing at several implications of your case study for the global carbon cycle.

Your manuscript is well written and presented. It hosts a wealth of information, but also draws extensively on previous work. Without doubt, your study fits the scope of the journal and has the potential to attract a broad audience of readers. Below I offer a number of comments and suggestions concerning both major sections and individual lines.

Section-Specific Comments:

- The title identifies "changing hydroclimate & landsliding" as the drivers of long term OC export. To be fair, you show next to no direct evidence of landslides (subaquatic mass transport aside). Would it be possible to estimate how much of the lake sediment may derive from other sources such as reworked fluvial or glacial material? The chosen slope threshold of 15° to define the catchments areas unduly biases your results to the relevance of landslides (see comment on line 568 below). If instead you can exclude all other sediment sources, there is no need to use a slope threshold, meaning that the inferred yields will be lower for a greater catchment area.

- The introduction concerns the erosion and sequestration of OC over various timescales. While readers may appreciate that extreme events such as earthquakes or rainstorms are important processes in this context, their contributions are bound to average out increasingly over longer time intervals. Perhaps you could state more clearly the timescale/s that is/are most relevant for OC export, and thus offer a more tangible definition of what you mean by "transience". Please clarify what you mean by "export" and "transfer", as you use these terms frequently. In this context, your lakes sit along a major plate boundary (Fig. 1), and sediment provenance might reveal the fractions of lake sediments derived from the Southern Alps as

opposed to the surrounding foreland. You might also want to discuss whether sediment trapping on the floodplains/fans between the mountains and the lakes is of relevant for your interpretations.

· Detailed volumetric reconstructions of sediment and carbon transfers: I understand that much of the sedimentary record discussed here comes from your previous publications (refs. 23, 25). The novel aspects here are the budgeting of sediment and OC masses and the palaeoclimatic inference for the two lakes. Please make sure to highlight more clearly these advances and avoid too much overlap with previous work.

· Extreme event forcing: consider adding a brief explanation of the earthquake cycle in your study area. Whether the purported rainstorm drivers were “extreme” is probably more difficult to estimate and deserves some discussion. Your global estimates of earthquake-induced landslides as a control on OC bio erosion (line 220) might be compromised by some of the limitations that you duly caution against in your introduction, i.e. the unconstrained extrapolation of short-term measurements. I also think that the point of hypothesizing about the role of extreme events is a bit moot, judging from the number of already existing studies that you cite (see detailed comments below).

· Climate drivers: in this section, it may be useful to acknowledge that climatic fluctuations are superimposed onto effects of the earthquake cycle. The text gives the impression that you can distinguish clearly between seismic and climatic (or weather) drivers in your sedimentary record, while these likely operate in concert. For example, picture a strong earthquake during a drier or wetter climate phase: would the geomorphic and sedimentary consequences be similar or different? Fig. 4 needs statistical support to highlight (anti-)correlations between sediment and OC yields and climate indices: this figure is more complex than what you describe in the text (see comments on line 239). Moreover, I found the section about estimating biospheric OC concentrations in rivers as a function of runoff (lines 266-298) awkward. You use power-law regression (line 272) to estimate OC concentrations that you translate into yields (Fig. 4) without disclosing how much material is trapped in the lake or lost via suspension. You then note that your “modelled yield is slightly higher than the lake derived estimate” (line 278), which you attribute to increased rainfall (and runoff?). Why not correct for this rainfall trend? Finally, you use the inferred increase in OC yields from the lake to argue for a commensurate increase in runoff (line 283). You seem to do this to satisfy primarily the model instead of testing the model on the data. The parsimonious explanation is that the model underfits the data. If so, interpreting the model predictions (line 289-298) may need commensurate care.

· Implications for the carbon cycle: in these final paragraphs, I would suggest revisiting your generalisations of how your case study reveals overall trends for “terrestrial ecosystems” or the “Earth system”. Why not address the partly implicit question regarding timescales: what do you take away, or recommend, in terms of capturing the “long-term” or “background” rate of sediment and OC fluxes? In other words, how do the contributions by extreme events smooth out, and is the lake record sufficiently long to offer informed statements about long-term trends?

· In both the Methods section and the main text, please explain more rigorously all errors where necessary: your conclusions hinge on how these errors are defined and how they propagate. Please distinguish more clearly errors in the data from errors in the models. You seem to mix methods of frequentist and Bayesian inference, and should be aware that their resulting errors have quite contrasting interpretations. Maybe consider a supplementary figure showing the workflow of data input, model extrapolation, and accumulated errors.

Line-Specific Comments:

21: “when organic carbon accumulates in sediments” - In which sediments? You mention “export of carbon from terrestrial ecosystems” at the start of this sentence, so I assume that these sediments are aquatic or marine?

22: “does not capture how transient changes in hydroclimate, and abrupt increases in erosion during extreme events” - Not sure if this is fully correct. Quite a number of studies have estimated carbon exports following extreme events, as you will be well aware of.

24: What are “4D reconstructions”?

27: “yields from both lakes show that earthquake- and storm-induced landslides” - This link between lakes and landslides may not be clear to all readers. Did the landslides drop into the lakes?

29: “Between landsliding episodes, carbon export increased up to two times during centuries with a wetter reconstructed climate.” - This reads as if, during a wetter climate, landslides were less active or prominent?

31: “active tectonics can dramatically increase organic carbon export from steep mountain forests, while a wetter climate can further increase carbon transfer” - Several questions come to mind here: “active tectonics” is earthquakes? If both these and a wetter climate increase OC export, what is responsible for the background rates? How can you tell apart the contributions of earthquakes during a wetter climate? And what is the difference between OC “export” and “transfer” here?

39: “to million-year timescales” is not mentioned in the abstract and would lend a completely different perspective to your sedimentary record.

39: Explain “canonical views”.

45: “-47-72 Mt.C.yr⁻¹” - Please keep notation tidy: in this sentence, you use three different variants, i.e. a range (is the first symbol a “minus”?); an asymmetric +/- error, and an approximate value (“~”). You may want to mention that all these are current estimates. I think that you can also safely drop the dots between the physical units.

45: Overuse of “globally” in this sentence. Consider mentioning this once.

52: “to the ocean” only? What about inland lakes?

54: “quantifications” -> “estimates”. Please clarify what you mean by “export from landscapes” here; your sedimentary record, for example, is still in a landscape.

56: “that span short timeframes of <101 years” -> “capturing less than a decade”?

57: “spatial regression models that assume apparent spatial patterns also inform temporal controls” - This sounds a bit vague. Please elaborate.

59: “These studies” refers to the previous sentence? The references you list here concern other research.

62: "Evaluating whether these feedbacks operate through time" - Sounds a bit trivial given the many processes you introduced so far.

64: "geomorphic and sedimentary processes play out over millennia" - To what consequence?

64: "transient periods of adjustment to climate change" - Depends on how you define "transience". Climate change has a clear temporal context, at least regarding contemporary atmospheric warming.

70: "secular change over timescales of 100s to 1000s of years" - So at which timescales do we stop speaking about "transience"?

71: "Despite their rarity, earthquakes and large storms" - Depends on the magnitude of these events. Earthquakes occur every day.

81: "global datasets of river yields" may need some reference(s).

83: "high density" of?

84: "over the entire lake basin" - Seems out of sequence: you have hardly mentioned the lake so far.

88: "natural experiment" -> "archive"? It is surely not an experiment, as you cannot repeat it under controlled conditions.

89: "demonstrate links between tectonic and climatic forcing" - You did not mention these links in the abstract.

91: "runoff variability" driven by climatic changes?

91: "from terrestrial ecosystems in the Southern Alps" may be a bit generous if you assume that your 60-km² study area is thus representative.

94: Figure 1: The isobaths are nice, though reveal little about the overall lake depths. Consider adding a maximum depth maybe. Explain why the northeastern parts of Lake Paringa have no catchment area.

104: "record" -> "records".

109: "are" -> "were".

111: "storm-driven landsliding" - How do you know that it was storms and landslides that sourced those sediments?

116: "2SE" - Spell out abbreviation the first time you use it; check grammar in this sentence.

117: "over the last 1023-1066 years" refers to which calendar year?

117: "6% of the total mass" - How come these percentages have no error bars?

118: Co-, post-, and interseismic phases may need a definition here.

124: Figure 2 would be stronger if it included controls on the absolute ages (14C samples, etc.), although I presume that most of the information presented here is contained in your previously published work (refs. 23, 25).

130: "1717 CE earthquake cycle" - How can one year be a cycle?

136: "SS" - How much bedload would those lakes be capable of trapping? Perhaps comment more on this fraction.

136: "phases" -> "deposits"?

140: "df=1, t=25, p<0.05" - These and other test statistics are useful only if you describe or mention at least the underlying test.

146: "from" -> "into"?

149: "most likely caused by storm-induced landsliding" - see comment on line 111. Given the amount of rain in the western Southern Alps, I find it curious to see so little evidence of storm-induced landsliding in your lake records.

158: "precision" - Yes, but what about accuracy?

161: "95% Highest Probability Density Function" - Do you mean a 95% highest density interval? While I welcome the use of intervals, this approach is somewhat at odds with that of the more limited use of statistical testing for differences in the means (e.g. line 140).

163: "then decayed to values within uncertainty" - Meaning the estimates are then indistinguishable with 95% probability? Please elaborate.

168: Figure 3 should show errors on the calendar ages.

181: "This demonstrates that lake stratigraphic records can preserve quantitative records of yields from mountain catchments that bridge the gap between instrumental and millennial timescales." - Agreed, but hardly anything novel?

179: "agree well with independent denudation rate estimates from cosmogenic nuclides" - It might be useful to add these estimates to Figure 3.

195: "shows" -> "show".

201: "extrapolating yields derived from short-term instrumental records over longer timeframes if they are representative of mountain catchments more generally" - Comes back to my initial point about the introductory section about timescales.

206: "Hence, we hypothesize that extreme event driven landsliding may play an important role in driving transients of SS, OCbio and OCpetro yield from catchments" - Perhaps "hypothesize" is the wrong phrase here, as you cite numerous studies with the necessary evidence.

208: "It also explains why SS [...] from the 32,000 km² Narayani R. catchment draining the Nepalese Himalaya" - Undue breaks in logical flow line the rest of this paragraph: you begin by "explaining" by referring to the previous hypothesis? Without any context, the example of the Nepalese river comes out of the blue and may be difficult to understand without any further details.

212: "typhoon" - Do you mean "monsoon"?

213-216: Reads more like introductory material. If this is meant to generalise your findings beyond your study area, you may wish to highlight also the key differences (instead of just some similarities) to other catchments of this size in Oceania.

217: "Together, this supports our hypothesis" - See comment on line 208. You seem to state a hypothesis and use this to confirm observations to then support your hypothesis. Avoid this circular reasoning and perhaps rephrase to: "Overall, our data support the hypothesis that extreme event driven landsliding causes substantial transient increases in SS and OCbio export that stand out in sedimentary archives formed over centuries to millennia."

225: "biome-based classification of vegetation and soil carbon stocks" needs a supporting reference.

227: Delete "annual"; it is part of the unit yield at the end of the sentence.

228: "yield" should be "flux"? At least, you use this term in the following sentence.

228: "the estimate is uncertain because events may be missing from the EQIL database" - and because the depth of erosion and local OC concentrations are probably even more uncertain.

231: "Despite the limitations we conclude" - You may want to offer some objective error estimates before concluding.

238: "opportunity to assess whether climate variability is linked to sediment and carbon transfers" - This would also be a good opportunity to frame a hypothesis before interpreting the data. What do you mean by "transfer": the following sentence refers to "export"?

239: "significant difference" - Might need a statistical treatment. Sediment fluxes in Figure 4b seem indistinguishable within error in the first (from the right; at around 1000 CE) and last phase shown. Yet, the Fiord Precip Index (Figure 4d) differs in that it steadily increased in the last phase, while it remained roughly constant in the first. Similarly, the second and third phase of biospheric OC yields (Figure 4a) seem to be indistinguishable from each other, whereas the SOI (red line, Figure 4f) was mostly negative during the second, and mostly positive during the third phase. Some more formal time-series analysis may be helpful here.

240: "In Lake Mapourika, the shorter length of the record precludes robust comparison" - Unclear. How much shorter is the record and why is this still sufficient then for drawing conclusions about differences in sediment and OC yields?

246-248: Errors needs specification; please note that the errors of low and high yield phases overlap. How well can you distinguish these mean rates from each other?

249: "This suggests that runoff" - This proposed link may need elaboration. How does SHWW connect to precipitation, and how does this connect to runoff in your study area?

250: "during time periods between large earthquakes and storms" - Why "between"? Should those climatic fluctuations not be superimposed onto the earthquake cycle?

251: "confirm hypotheses from instrumental studies that use spatial variations in climate to show runoff controls OCbio export" - This statement is a bit vague. Please be more specific.

254: "Fiord" Precipitation Index needs explanation.

268: "fit to empirical data from mountain river catchments globally" - Is this fit thus prone to the limitations that you cautioned against earlier, i.e. interpolating short-term measurements?

270: "described by a function of" -> "estimated from". It might pay off to add that the equation in line 272 estimates only the average river particulate biospheric OC concentration.

273: "pre-factor that is related to the slope angle of the catchment" - Unclear. Why is this coefficient only related to slope?

274: "Assuming $\alpha = 0.052$ " - Alpha is not dimensionless; please add units. Both alpha and gamma have errors, which should enter the following estimates.

276: Explain "CI".

281: "measured" -> "inferred".

283: "can be explained by the erosion model" - Do you mean the equation in line 272? Why do you require the data to the fit the model? If anything, it should be the other way 'round. You should contemplate that the simple model in line 272 may fail to capture all variance in the data. Hysteresis would be a first effect to consider when looking at relationships between runoff, discharge, and material concentrations in rivers.

295: "doesn't" -> "does not"; "for the predicted increase in severity of the largest flow events ... because it scales non-linearly with runoff" - Do you mean that equation in line 272 is non-linear? But why then are the largest flow events a problem? Either the model is capable of accurate prediction or it is not. Note that you refer to "erosion" in line 296, whereas the equation estimates concentrations.

289: "Building on this assessment of the lake records using the runoff-driven model" - This reads as if you assess the data based on your model, whereas the way that statistical inference works is vice versa.

304: "how orogenesis, erosion and weathering contribute to the carbon cycle" - Your study addresses none of these likely contributors directly.

310: "fluxes are mediated by hydroclimatic variability" - What do you mean by "meditated" here? Please see comments on line 239.

312: "climate change from the Little Ice Age" - Please remind readers about the timing of this in the Southern hemisphere, and consider showing this interval in Figure 4.

314: "Our data shows that erosion of OCbio could act as a negative feedback in the Earth System" - Maybe tone this down a bit. You generalise your findings from sediments in two lakes in New Zealand to erosion in the mountain catchments upstream, and then the Earth System? Maybe remind readers which data or evidence you refer to here.

315: "which" refers to?

316: "Wetter

317 "climates associated with warmth can enhance erosion of OCbio from terrestrial ecosystems, promote its transfer into long-term sedimentary storage, and drawdown CO₂ from the atmosphere (Fig. 4)" - Not sure whether Fig. 4 shows this clearly, let alone whether it is prudent to generalise your findings; see previous comment.

321: "is thought to be typical" sounds vague and as if it reflects models fit to data?

480-494: Please explain the nature of all errors cited. This consistency is essential as you base a lot of your conclusions on quantitative estimates, in which these errors will accumulate. Some errors seem to be standard deviations about the mean, other confidence intervals, and yet others credible intervals. If combining these, please consider their differing interpretations.

517: "sill that separates the Windbag Basin from the Hall Basin" - First time you mention the Hall Basin without much context.

541: "derive" -> "estimate". The durations of these phases thus should have age errors.

554: Index "n" is the number of voxels?

564: "$\pm 5\%$" makes no sense mathematically.

564: "uncertainty was propagated through" - How?

568: "A is the catchment area with hillslope angle above 15°" - I understand that you use this threshold to include landslide-prone hillslopes only. Why would landslides not occur beyond this threshold? Debris flows might enter the lake(s) at lower terrain inclinations. You might unnecessarily bias your sediment and OC yields by assuming that only landslide input is relevant. It may be a fair assumption that soil erosion, surface runoff, etc. produce negligible input, but you cannot infer the

relevance of landslides from your data if you admit little else in terms of other geomorphic processes.

615: "from two range front catchments near Lake Mapourika with the same physiographic characteristics as Potters Creek" - This reads a bit vague given the previous efforts that have tried to link denudation rates from cosmogenic nuclides to all sorts of terrain characteristic. Do these sampled catchments also contain parts of the foreland? Maybe show these sample locations in Fig. 1?

624: "similar physiographic setting (New Zealand and Taiwan)" - See previous comment about similar physiographic setting. Maybe limit this to data from New Zealand?

629: "there is no comparable data for validating the Lake Paringa SS yields" - Assuming that the rates derived from cosmogenic nuclide concentrations reflect the true rates? It might be sufficient to compile the available data on erosion and denudation here: each of the methods has its drawbacks, but they constrain the range of rates quite well.

636: "Our estimates of OCbio yields for a given SS yield agree well with global values" - What do you mean by "agree well"?

640: "This trend is likely compounded" - Which trend? Do you mean "confounded" or "compounded"? If you mean the latter, what are the other components other than high OC stocks in soils?

646: "Seal et al.,33" - Reference appears to be incomplete in the list. Please check.

651: "window" -> "interval"?

651: What is the "typical return interval of earthquakes"?

658: "assess" - "use"?

661: "may represent over- or under- estimates" -> "be misestimates". So why then would the sum of these be any more accurate than individual events? Please elaborate.

Fig. S7: Caption should explain dotted lines and green box.

Fig. S8: Regression equation is valid only for mean Fpetro. How do you deal with the variance when you use this model for predictions?

Fig. S9: Please explain the two lines. Why do the global data have no errors?

Communications Earth & Environment is committed to improving transparency in authorship. As part of our efforts in this direction, we are now requesting that all authors identified as 'corresponding author' create and link their Open Researcher and Contributor Identifier (ORCID) with their account on the Manuscript Tracking System prior to acceptance. ORCID helps the scientific community achieve unambiguous attribution of all scholarly contributions. You can create and link your ORCID from the home page of the Manuscript Tracking System by clicking on 'Modify my Springer Nature account' and following the instructions in the link below. Please also inform all co-authors that they can add their ORCIDs to their accounts and that they must do so prior to acceptance.

Version 1:

Decision Letter:

Dear Dr Howarth,

Your revised manuscript titled "Long-term carbon export from mountain forests driven by hydroclimate and extreme event-driven landsliding" has now been seen by our original reviewer #2, whose comments appear below. In light of their advice we are delighted to say that we are happy, in principle, to publish a suitably revised version in Communications Earth & Environment, provided you address all remaining discussion points and clarify all methodological details.

We therefore invite you to revise your paper one last time to address the remaining concerns of our reviewer #2. At the same time we ask that you edit your manuscript to comply with our format requirements and to maximise the accessibility and therefore the impact of your work.

EDITORIAL REQUESTS:

*****Please take care to match our formatting and policy requirements. We will check revised manuscript and return manuscripts that do not comply. Such requests will lead to delays. *****

SUBMISSION INFORMATION:

OPEN ACCESS:

Communications Earth & Environment is a fully open access journal. Articles are made freely accessible on publication. For further information about article processing charges, open access funding, and advice and support from Nature Research, please visit <https://www.nature.com/commsenv/open-access>

Link Redacted

Best regards,

Adam Switzer, PhD
Editorial Board Member
Communications Earth & Environment
orcid.org/0000-0002-4352-7852

Carolina Ortiz Guerrero, Ph.D.
Associate Editor,
Communications Earth & Environment
Consulting Editor, Communications Sustainability

REVIEWERS' COMMENTS:

Reviewer #2 (Remarks to the Author):

Review of manuscript #COMMSENV-24-2345A: "Long-term carbon export from mountain forests driven by hydroclimate and extreme event-driven landsliding" submitted by J. Howarth and colleagues

Dear Authors,

Thank you for your detailed reply letter concerning the remarks and suggestions of the previous round of reviews. Your answers have clarified most of my initial queries, and I appreciate how thoroughly you revised the original manuscript. Below I list a number of comments that you may want to look at once more. These refer mainly to how you handle errors and how this might affect your discussion about the global relevance of your study. Other than that, thanks again for this sneak preview: I have learned a lot about how one can estimate sediment and OC yields from the western Southern Alps.

Line-specific comments (refer to track changes version):

2: "extreme event-driven landsliding" – I maintain my original comment that you show very little evidence of landslides.

70: "has been revealed using spatial regression models" – Consider replacing "revealed" by "estimated" for reasons that you spell out in this new paragraph. Again, I have yet to see "spatial" regression models. The "space for time" substitution you mention in line 78 is not necessarily a spatial regression model in the geostatistical sense. Try avoiding this ambiguity.

79: "scaling between variables spans an order of magnitude" – Please specify these variables.

171: How do you define or estimate the "long-term average" and why? You put a lot of emphasis on the importance of decadal to millennial intervals in sedimentary records, and hence might also want to establish what you feel is the necessary background for reference.

173: "average SS yield over the record duration was ..." - See previous comment: is this the "long-term average"?

190: What are the "timescales relevant to the geological carbon cycle" and how can you tell.

193: "Our approach to reconstructing sediment and carbon yields from lake sediments could be deployed globally" – How would that work? It seems quite an effort to study what you did for a couple of lakes.

194: "over timeframes relevant to the carbon cycle" – See previous comment.

197: "ours" → "our".

234: "These small, highstanding, and forested catchments have OC_bio yields that are, on average, four times higher than yields from other physiographic settings" – Does this comparison rely on short-term yields that you caution extrapolating with each other?

237: "our findings, while based on only two catchments, may have global relevance because ... SS and OC_bio yields derived over years to a decade from river gauging data may underestimate long-term yields for catchments that are a significant source of OC_bio export to the global ocean" – Suggest revisiting this statement for two reasons. (1) Other studies in similar settings might differ concerning their earthquake cycles or length of record, and accordingly so the mismatch between shorter- and long-term rate estimates. It is a valid point, though, to stress the discrepancy between gauge readings and volumetric estimates of lake fills. (2) How are your catchments "significant source of OC_bio export to the global ocean" with much of their output trapped in lakes?

238: "suggests" → "suggest".

242: "the spatial scale of event-driven landsliding is low relative to the total catchment area, limiting the magnitude of transient increases in OC_bio export caused by extreme-event driven landsliding" – This may need a supporting reference or two. One could think that, in these settings, the mismatch between short-term readings and long-term effect of rare earthquakes or rainstorms could be even higher.

247: "showed a negligible increase in SS or OC_bio yields following the 2016 Mw7.8 Gorka earthquake that triggered extensive landsliding" – Perhaps it would be useful to elaborate briefly why. Before the background of discussing a more regional, if not global, relevance, this particular study (ref. 18) seems to limit the notion of wider applicability of yours. Quite the contrary, this difference motivates a closer look at the underlying controls and promises some interesting future work.

249: "only impacted the eastern part of the catchment and represented only a modest increase over annual monsoon driven rates of landsliding" – And this scenario of localized landsliding is something that you can exclude for the western Southern Alps? My understanding is that the Gorkha earthquakes still triggered tens of thousands of landslides.

261: "while temporal variations in EQIL rate may not be captured by the 50-year timeframe" – I am not even convinced that the estimated mean EQIL rate is reliable, given that the catalogue you use is heavily biased by recent, and more detailed, studies.

276: "to assess the hypothesis derived from space for time substitution experiments that temporal variability in hydroclimate drives changes in sediment and carbon yields" – Note that the space-for-time approach is a hypothesis on its own. To test whether variable hydroclimate is a possible driver, you may need to demonstrate that sediment and OC yields remain unchanged without considering hydroclimate.

303: "our findings also confirm hypotheses from space for time substitution experiments that use OC_bio yield from river gauging and spatial gradients in climate" – Not sure whether you can "confirm" these hypotheses at all: they would be fact eventually. How is the space-for-time substitution for gauging data not prone to a possible short-term bias that you cautioned against earlier?

316: Coefficient alpha has incorrect unit: this must also depend on exponent gamma. If the equation in line 315 is a regression model, it misses an error term. At the very least, the OC yields in this and the following paragraph should carry the estimated regression error.

318: "we assume $\alpha = 0.052$ and $\gamma = 1.37$, which reflects the Lake Paringa catchment" – Why assume? Do you not obtain this via regression? And what about the catchment feeding Lake Mapourika?

320: "mean annual runoff of 3848 ± 440 (95% CI) calculated from the Lake Paringa catchment rain gauge" – Runoff needs unit. How long is this record and how does it safeguard against the problem of unduly extrapolating over longer time

periods?

356: "Our reconstructions support the hypothesis based on global scale space for time substitution experiments on modern observations" – See previous comments. I would be more careful in discussing how well you can extrapolate your local findings for reasons that you clearly spell out.

912/Fig. 4: It might be useful to add to the boxplots of SSY and OC which of the four intervals you can tell apart from each other within error. In other words, how well can you establish that yields in phases 1 and 4 of SSY and phases 2 and 3 of OC are different?

22 December, 2024

To the reviewers,

We thank two anonymous reviewers for their time and effort so far in assessing and improving our paper, "Long-term carbon export from mountain forests driven by hydroclimate and extreme event-driven landsliding." Both reviewers provided positive reviews that highlighted that our work was innovative and appropriate for publication in *Communications in Earth and Environment*.

We are grateful for the detailed and constructive feedback that has helped us improve the content and clarity of the manuscript. We have made extensive revisions to the manuscript in response to the comments. These changes are itemised below in response to each comment. Reference line numbers in the new text are provided so changes can be easily identified. We also provide detailed explanations and rebuttals to some of the more general comments.

Kind regards,

Jamie Howarth.

REVIEWER COMMENTS:

Reviewer #1 (Remarks to the Author):

This is a review report for the manuscript, entitled: Long term carbon export from mountain forests driven by changing hydroclimate and landsliding, by Dr. Howarth et al. This study took many cores in Lake Paringa and Mapourika, New Zealand and reconstructed the SS, OC_{bio}, and OC_{petro} export for millennial timescales. They found that Landsliding triggered by earthquakes or rainstorms plays a dominant role in OC export, which can be an additional CO₂ drawdown, even overwhelms silicate weathering. Moreover, after episodic landsliding, the elevated OC_{bio} export which can persist for a long time (at least 4-10 yr, even longer) is also significant negative feedback. The former argument has been discussed/published/validated for some recent studies. The later argument (prolonged elevated OC_{bio} export) is very innovative.

Thank you for these positive comments, we are pleased the novel aspects of our study were recognised.

I have two concerns before suggesting to the next publication stage.

The first one is that the sequence of OC_{bio} from landslides to transport to burial (lake sediments) has been demonstrated by authors. The signal or amount of OC_{bio} in this case may be special due to the short pathway. The longer pathway or travel time enhances OC_{bio} decomposition. With the increase of travel time, the increasing OC_{bio} decomposition may hinder the negative feedback. At

global scale, I am curious how to take this into considering global riverine OC_{bio} export [Line: 220-233].

We agree that our findings may not apply to the largest global rivers draining large continental areas, such as the Amazon or Ganges, where OC_{bio} may be remineralised as it traverses through long floodplains. However, this does not undermine the importance of our findings on a global scale. We argue that our catchments are generally representative of the relatively small <10³ km² catchments that dominate the high-standing islands of Oceania, where OC_{bio} yields from rivers are nearly a factor of four higher than other measured rivers globally (Hilton and West, 2020). It is well established that these mountain islands deliver sediment to the ocean effectively and in fact dominate global sediment fluxes (Milliman and Sytiksi, 1992; Larsen et al., 2014). Indeed, they are thought to contribute a third of OC_{bio} flux from land to the global Ocean (Lyons et al., 2002). We have added this specificity to lines 236 and 264 in the main text to clarify the general relevance of our work better. Consequently, while we agree that our work represents a single case study, it is important because it captures the behaviour of high-standing, forested catchments with a relatively short source-to-sink distance, which are important for global OC_{bio} export by river to the global ocean.

You raise a valid point about our global estimate for the influence of earthquake-induced landslide-driven OC_{bio} export. We agree that the transport pathway will be an important modifier on the amount of OC_{bio} mobilised by earthquakes that make it to the ocean. When using the global earthquake triggered landslide database, our approach is simple – it just quantifies the mass of OC_{bio} mobilised on hillslopes by earthquake induced landsliding. We cannot directly track the ultimate fate of this material and this remains a future research priority. However, we note that it shows why our lake study is so useful, because there we can track the transport of OC_{bio} after several large earthquakes over a thousand years. We have now specified both the need for caution when interpreting this global earthquake triggered landsliding upscaling result, but also its importance given the location of the earthquakes in the inventory in lines 256 and 262.

Hilton, R. G., and West, A. J., 2020, Mountains, erosion and the carbon cycle: *Nature Reviews Earth & Environment*, v. 1, no. 6, p. 284-299.

Lyons, W. B., Nezat, C. A., Carey, A. E., and Hicks, D. M., 2002, Organic carbon fluxes to the ocean from high-standing islands: *Geology*, v. 30, no. 5, p. 443-446.

Milliman, J. D., and Syvitski, J. P. M., 1992, Geomorphic/Tectonic Control of Sediment Discharge to the Ocean: The Importance of Small Mountainous Rivers: *The Journal of Geology*, v. 100, no. 5, p. 525-544.

Milliman, J. D., 2011, River discharge to the coastal ocean a global synthesis, in Farnsworth, K. L., ed.: *Cambridge* ;, Cambridge University Press.

The second one is Fig.4:

As core and dating data showed an extreme earthquake occurred in ~1717CE and buried a lot of OC_{bio} (Fig. 2). I just wonder why this can't be find in Fig. 4(a)? It looks like a long-term average. According to Fig.3, I think you can have more than 10 dots for Fig. 4(a). Even the OC_{bio} exports are different in the two lakes, I think their changes are consistent.

Here we only show the inter-seismic periods. That is why in this figure you cannot see the earthquake event. We have clarified this in the figure caption.

We do this to explore how the elevated extreme event driven SS and OC_{bio} yields are superimposed on longer term variability driven by climatic change, in this case hydrometeorology. When the studies catchments are responding to these extreme events, they are unlikely to be sensitive to longer-term changes in hydroclimate. Consequently, we only compare interseismic rates, when the catchments are not responding to extreme events to the hydroclimate proxies, to examine whether variability in OC_{bio} yields over >10² year timescales is related to changes in climate. We have clarified this reasoning between lines 290 and 294.

Minor comments:

What do the lines in supplementary Fig. 9 stand for? In the Fig. 9(a), is it a regressive line? If they are regressive line, please provide the r² and p-value.

You are correct, the line is the regression from Galy et al., (2015). We have added the regression equation, p-value and R² to Supplementary Figure 9.

Reviewer #2 (Remarks to the Author):

Review of COMMSENV-24-2345-T: "Long term carbon export from mountain forests driven by changing hydroclimate & landsliding" by J. Howarth and colleagues

Dear Authors,

In this contribution you estimate the flushing of sediment and organic carbon (OC) from small catchments draining the western Southern Alps of New Zealand. You argue that rainstorms and strong earthquakes are the main processes responsible for episodically eroding densely forested hillslopes, and thus delivering large amounts of OC to two foreland lakes that serve as sediment traps and archive such pulses. Drawing on your previous research, in which you identified several cycles of sedimentation following major earthquakes, you offer here mass estimates of material deposition during co-, post-, and interseismic intervals over the past millennium. You argue that strong earthquakes and rainstorm-induced episodes of sediment influx to the lakes have average OC yields that outweigh those during longer periods of relative geomorphic quiescence. You also propose that palaeoclimatic proxies of regional precipitation or wetness help to explain variations in the inferred sediment and OC yields. You conclude by pointing at several implications of your case study for the global carbon cycle.

Your manuscript is well written and presented. It hosts a wealth of information, but also draws extensively on previous work. Without doubt, your study fits the scope of the journal and has the potential to attract a broad audience of readers. Below I offer a number of comments and suggestions concerning both major sections and individual lines.

Thank you for recognising the significance of the new work here.

Section-Specific Comments:

· The title identifies "changing hydroclimate & landsliding" as the drivers of long term OC export. To be fair, you show next to no direct evidence of landslides (subaquatic mass transport aside). Would

it be possible to estimate how much of the lake sediment may derive from other sources such as reworked fluvial or glacial material? The chosen slope threshold of 15° to define the catchments areas unduly biases your results to the relevance of landslides (see comment on line 568 below). If instead you can exclude all other sediment sources, there is no need to use a slope threshold, meaning that the inferred yields will be lower for a greater catchment area.

We agree with the reviewer that the title needs to be more specific to highlight better the contribution the research makes. As such, we have changed the title to 'Long-term carbon export from mountain forests driven by hydroclimate and extreme event-driven landsliding' because we feel this better reflects what our sedimentary records show i.e. pulsed sediment and carbon export from extreme event-driven landsliding, rather than landsliding more generally, which we know dominates sediment yields from the Southern Alps in between the extreme events (Hovius et al., 1997).

We disagree that the thresholding of the catchment area to slopes above 15 degrees bias our sediment and carbon yields. Previous studies on sediment provenance from Lake Paringa have shown that the vast majority (>90%) of sediment is sourced from elevations (>200 m) in the catchment that have slopes greater than 15 degrees (Figure 1, Wang et al. 2020). It also makes our lake-derived yields more comparable to global river datasets for mountain rivers that tend to measure sediment yield from the range front (Hilton et al., (2012). We now mention this specifically in the methods section between lines 370 and 371.

Fig. 1 The elevation source of organic matter in Lake Paringa during the inter- and postseismic periods (from Wang et al., 2020).

(A) The blue and red lines show the elevation distributions of eroded soil for the whole record (Fig. 3C) during the inter- and postseismic period, respectively. The dashed lines show the 16th, 50th, and 84th percentiles of the distributions. (B) The gray line shows the elevation distribution of the alpine catchment (the west-flowing catchment of the Windbag River) with 16th, 50th, and 84th percentiles of the distribution as dashed lines. The black line is the distribution of slopes larger than 20° at alpine catchment. The frequency data of (A) and (B) have been binned in 25-m vertical intervals.

Hilton, R. G., Galy, A., Hovius, N., Kao, S.-J., Horng, M.-J., and Chen, H., 2012, Climatic and geomorphic controls on the erosion of terrestrial biomass from subtropical mountain forest: *Global Biogeochemical Cycles*, v. 26, no. 3.

Hovius, N., Stark, C. P., and Allen, P. A., 1997, Sediment flux from a mountain belt derived by landslide mapping: *Geology*, v. 25, no. 3, p. 231-234.

Wang, J., Howarth, J. D., McClymont, E. L., Densmore, A. L., Fitzsimons, S. J., Croissant, T., Gröcke, D. R., West, M. D., Harvey, E. L., Frith, N. V., Garnett, M. H., and Hilton, R. G., 2020, Long-term patterns of hillslope erosion by earthquake-induced landslides shape mountain landscapes: *Science Advances*, v. 6, no. 23, p. eaaz6446.

The introduction concerns the erosion and sequestration of OC over various timescales. While readers may appreciate that extreme events such as earthquakes or rainstorms are important processes in this context, their contributions are bound to average out increasingly over longer time intervals. Perhaps you could state more clearly the timescale/s that is/are most relevant for OC export, and thus offer a more tangible definition of what you mean by “transience”. Please clarify what you mean by “export” and “transfer”, as you use these terms frequently

We are grateful for the reviewer's comments on the introduction and have edited the text to highlight the timescales most relevant to the role of OC_{bio} export and burial in the global carbon cycles.

While it is true that the contribution of extreme events to perturbations in sediment and carbon yields could appear to average out over the long term, the key point here is that the vast majority of estimates of these yields come from short-term (a few years to a decade) river monitoring datasets. Therefore, existing global estimates do not account for the contribution of extreme events or secular changes in climate. We have modified the text between lines 43 and 81 to make this point overt. Transience is now clearly defined as temporal variability in yields, and we define the timescales for extreme landslide events and climate change between lines 56 and 59. We agree that export and transfer needed to be defined better on first use. We now define export explicitly before its first use on line X; and we no longer use transfer to avoid ambiguity.

In this context, your lakes sit along a major plate boundary (Fig. 1), and sediment provenance might reveal the fractions of lake sediments derived from the Southern Alps as opposed to the surrounding foreland. You might also want to discuss whether sediment trapping on the floodplains/fans between the mountains and the lakes is of relevant for your interpretations.

These elements have been dealt with in previous publications e.g. Wang et al., (2020). We now include specific mention of this in the methods between lines 370 and 371.

Detailed volumetric reconstructions of sediment and carbon transfers: I understand that much of the sedimentary record discussed here comes from your previous publications (refs. 23, 25). The novel aspects here are the budgeting of sediment and OC masses and the palaeoclimatic inference for the two lakes. Please make sure to highlight more clearly these advances and avoid too much overlap with previous work.

Where we leverage existing work, we briefly summarise previous findings and provide citations that clearly show this is previous work. We take this approach to ensure the reader can understand the narrative without having to unnecessarily delve into previous literature. All the findings reported in the results are new and based on our detailed sediment budgeting that has been underpinned by extensive new analysis of 10s of sediment cores and detailed sedimentology cross-correlation methods.

· Extreme event forcing: consider adding a brief explanation of the earthquake cycle in your study area. Whether the purported rainstorm drivers were “extreme” is probably more difficult to estimate and deserves some discussion. Your global estimates of earthquake-induced landslides as a control on OC_{bio} erosion (line 220) might be compromised by some of the limitations that you duly caution against in your introduction, i.e. the unconstrained extrapolation of short-term measurements. I also think that the point of hypothesizing about the role of extreme events is a bit moot, judging from the number of already existing studies that you cite (see detailed comments below).

We have provided a brief summary of the earthquake cycle of the Alpine Fault between lines 87 and 88 in the introduction and expanded on this in the methods section between lines 356 and 358. We cannot report on the magnitude of the rainfall that triggered landslide-induced sediment pulses because this is unknown. However, the magnitude of the increase in sediment yields shows they were extreme events from that perspective.

While we agree that our global estimation of OC_{bio} mobilisation from earthquake induced landsliding would be improved with a longer record, it at least spans multiple decades, rather than the years of spot measurements used to quantify OC_{bio} export from river gauging. We note that such an analysis has not been attempted before, and it adds novelty to the findings.

On the advice of reviewer 1 we have restructured the section between lines 213 and 241, so the comment about the extreme event hypothesis is now redundant.

· Climate drivers: in this section, it may be useful to acknowledge that climatic fluctuations are superimposed onto effects of the earthquake cycle. The text gives the impression that you can distinguish clearly between seismic and climatic (or weather) drivers in your sedimentary record, while these likely operate in concert. For example, picture a strong earthquake during a drier or wetter climate phase: would the geomorphic and sedimentary consequences be similar or different? Fig. 4 needs statistical support to highlight (anti-)correlations between sediment and OC yields and climate indices: this figure is more complex than what you describe in the text (see comments on line 239).

We are grateful to the reviewer for their thoughts on this. We agree that transient increases in sediment and carbon yields following extreme events will be modulated by climate among a range of other variables (e.g., the density of earthquake-induced landsliding). However, our approach does allow the influence of secular changes in climate to be isolated from the influence of extreme event-driven landsliding.

In summary, our approach is simple: when comparing our sediment yield estimates with climatic change, we remove the intervals when the catchment responds to the widespread earthquake-induced landslides. This approach allows the relationship between inter-seismic sediment yields to be compared with reconstructed hydroclimate i.e. for the time periods when the Lake Paringa catchment is not responding to earthquake-induced landsliding.

As we only have four inter-seismic sediment and carbon yield estimates from Lake Paringa it is not appropriate to conduct a formal timeseries analysis on this data because n is too low. The best we can do is demonstrate that there is a statistically significant difference between interseismic sediment yields (Chi squared test that shows the distributions are significantly different). Besides the vastly different temporal resolution of the paleoclimate reconstructions make formal time series analysis challenging. However, we have added a more nuanced description of the potential relationship between lines 273 and 293, but still maintain there is a general relationship between reconstructed hydroclimate and sediment and carbon yields.

Moreover, I found the section about estimating biospheric OC concentrations in rivers as a function of runoff (lines 266-298) awkward. You use power-law regression (line 272) to estimate OC concentrations that you translate into yields (Fig. 4) without disclosing how much material is trapped in the lake or lost via suspension. You then note that your “modelled yield is slightly higher than the lake derived estimate” (line 278), which you attribute to increased rainfall (and runoff?). Why not correct for this rainfall trend? Finally, you use the inferred increase in OC yields from the lake to argue for a commensurate increase in runoff (line 283). You seem to do this to satisfy primarily the model instead of testing the model on the data. The parsimonious explanation is that the model underfits the data. If so, interpreting the model predictions (line 289-298) may need commensurate care.

We thank the reviewer for their comments on this section and have made the following revisions based on their advice to improve the clarity of this section. Indeed, we have now computed the trapping efficiency for the lake, which is very high (92%). This allows a more direct comparison between modelled and reconstructed OC_{bio} yields. We have also corrected the runoff values used to compute the mean runoff for the trend of increasing rainfall. When this correction is completed, the modelled and reconstructed OC_{bio} yields agree within uncertainty, simplifying the narrative in this section. Thank you for spotting this. Further, our intention was not to validate the reconstructed yields with the model, but rather to show consistency between the reconstructed yields and a model derived using a space for time substitution and short-term yield data from rivers. We have modified the narrative in this section to make this clear. That’s for spotting the awkward nature of the original narrative.

The original purpose of calculating the runoff for the little ice age interval given the reconstructed OC_{bio} yields using the model was to sanity check the variability in runoff through time. However, on reflection we agree with the reviewer that this analysis is speculative because we don’t know how

runoff has varied through time, only precipitation, and so this analysis is no longer included.

· Implications for the carbon cycle: in these final paragraphs, I would suggest revisiting your generalisations of how your case study reveals overall trends for “terrestrial ecosystems” or the “Earth system”. Why not address the partly implicit question regarding timescales: what do you take away, or recommend, in terms of capturing the “long-term” or “background” rate of sediment and OC fluxes? In other words, how do the contributions by extreme events smooth out, and is the lake record sufficiently long to offer informed statements about long-term trends?

The reviewer raises a very good point here, which was only omitted in the submitted version due to the short manuscript length of the original target journal. We have now added a section between lines 163 and 199 that explores how wrong short-term estimates of sediment and carbon yields could be compared to the longer-term (millennial average) based on our reconstructed sediment and carbon yields.

· In both the Methods section and the main text, please explain more rigorously all errors where necessary: your conclusions hinge on how these errors are defined and how they propagate. Please distinguish more clearly errors in the data from errors in the models. You seem to mix methods of frequentist and Bayesian inference, and should be aware that their resulting errors have quite contrasting interpretations. Maybe consider a supplementary figure showing the workflow of data input, model extrapolation, and accumulated errors.

We appreciate the reviewer’s comments on the difference between Bayesian and frequentist approaches and, in particular, the difference in how their uncertainties are interpreted. We know the differences but feel our approach is defensible for two reasons. First, we use a Bayesian approach for deriving the age-depth models for the master cores from each lake because this is the best practice in Quaternary environmental reconstruction. Frequentist approaches to age-depth modelling (e.g., fitting spline) are no longer considered best practice. This leaves us with an invidious choice: apply a sub-optimal approach to age-depth modelling and be critiqued for not using best practice, or combine Bayesian and frequentist uncertainties in our error propagation. We have chosen the latter.

We have now provided statements on how uncertainty was propagated at each step through the methods to make it crystal clear how uncertainty was accounted for at each step and propagated (see additions between lines 418 and 495). However, we have not added another figure (we already have 9 supplementary figures) on uncertainty propagation because this is now clearly articulated in the methods.

Line-Specific Comments:

21: “when organic carbon accumulates in sediments” - In which sediments? You mention “export of carbon from terrestrial ecosystems” at the start of this sentence, so I assume that these sediment are aquatic or marine?

This comment is no longer relevant because we have had to re-write the abstract to make it fit the 150 word limit.

22: “does not capture how transient changes in hydroclimate, and abrupt increases in erosion

during extreme events” - Not sure if this is fully correct. Quite a number of studies have estimated carbon exports following extreme events, as you will be well aware of.

The full sentence states “Prior constraint has come from discrete sampling of modern rivers that does not capture how transient changes in hydroclimate, and abrupt increases in erosion during extreme events, **modify carbon export over decades or longer**.”. As there are very few river OCbio yield datasets that span decadal timeframes, we think this sentence as originally stated is factually correct.

24: What are “4D reconstructions”?

The reviewer has a fair point here because ‘4D reconstructions’ is not standard nomenclature, we were trying to capture the notion of volumetric constructions through time. However, for brevity we now simply state ‘volumetric reconstructions’.

27: “yields from both lakes show that earthquake- and storm-induced landslides” - This link between lakes and landslides may not be clear to all readers. Did the landslides drop into the lakes?

We have removed the words ‘from both lakes’ to avoid ambiguity (see lines 21 and 23).

29: “Between landsliding episodes, carbon export increased up to two times during centuries with a wetter reconstructed climate.” - This reads as if, during a wetter climate, landslides were less active or prominent?

We have modified the sentence to ‘Between extreme event-driven landsliding, carbon export...’ to avoid this ambiguity.

31: “active tectonics can dramatically increase organic carbon export from steep mountain forests, while a wetter climate can further increase carbon transfer” - Several questions come to mind here: “active tectonics” is earthquakes? If both these and a wetter climate increase OC export, what is responsible for the background rates? How can you tell apart the contributions of earthquakes during a wetter climate? And what is the difference between OC “export” and “transfer” here?

This sentence had a lot going on. The abstract has now been written to meet the journal's word count, so the sentence is no longer included.

39: “to million-year timescales” is not mentioned in the abstract and would lend a completely different perspective to your sedimentary record.

Given that, almost all constraints on OCbio export currently come from river datasets that span only a few samples over less than a year, we do not think this is an issue.

39: Explain “canonical views”.

We do not understand the query here. The canonical views of the carbon cycle are explained in the subsequent sentence.

45: “-47-72 Mt.C.yr-1” - Please keep notation tidy: in this sentence, you use three different variants, i.e. a range (is the first symbol a “minus”?); an asymmetric +/- error, and an approximate value (“~”).

You may want to mention that all these are current estimates. I think that you can also safely drop the dots between the physical units.

The variable notation for the original values were to preserve how they were reported in the publications from which they are derived. However, we take the reviewers points and have tidied the notation where possible so that only ranges are now reported.

45: Overuse of “globally” in this sentence. Consider mentioning this once.

This correction has been made.

52: “to the ocean” only? What about inland lakes?

Yes, to the ocean, no one has quantified the flux to inland lakes, and the relevance to the carbon cycle of OC_{bio} export requires long term burial in marine sediment.

54: “quantifications” -> “estimates”. Please clarify what you mean by “export from landscapes” here; your sedimentary record, for example, is still in a landscape.

We have made the adjustment and now specify terrestrial environments to be more specific, and note that many of the gauging stations for rivers used in previous estimates of OC_{bio} export are also still in the landscape, so using the lake as a gauging station is analogous to previous approaches.

56: “that span short timeframes of <101 years” -> “capturing less than a decade”?

We have made the change.

57: “spatial regression models that assume apparent spatial patterns also inform temporal controls” - This sounds a bit vague. Please elaborate.

We have modified the text to make it clear that we are referring to the practice of using space for time substitutions when datasets do not span sufficient time frames to assess temporal trends directly (see lines 74 and 79).

59: “These studies” refers to the previous sentence? The references you list here concern other research.

The text has been rewritten so that a separate sentence now articulates the general issues with using space for time substitutions to infer process controls (lines 74 and 76).

62: "Evaluating whether these feedbacks operate through time" - Sounds a bit trivial given the many processes you introduced so far.

64: "geomorphic and sedimentary processes play out over millennia" - To what consequence?

64: "transient periods of adjustment to climate change" - Depends on how you define "transience". Climate change has a clear temporal context, at least regarding contemporary atmospheric warming.

70: "secular change over timescales of 100s to 1000s of years" - So at which timescales do we stop speaking about "transience"?

71: "Despite their rarity, earthquakes and large storms" - Depends on the magnitude of these events. Earthquakes occur every day.

81: "global datasets of river yields" may need some reference(s).

Comments between for lines 62 to 81 are now redundant because this part of the introduction has been restructured and in doing so these comments have been addressed.

83: "high density" of?

This now reads 'A high density coring approach', which improved the clarity.

84: "over the entire lake basin" - Seems out of sequence: you have hardly mentioned the lake so far.

Fair point, this now reads 'accumulation in two lakes basins' to improve clarity.

88: "natural experiment" -> "archive"? It is surely not an experiment, as you cannot repeat it under controlled conditions.

We have made this change.

89: "demonstrate links between tectonic and climatic forcing" - You did not mention these links in the abstract.

This comment is no longer relevant because we have restructured the abstract.

91: "runoff variability" driven by climatic changes?

We have made this change.

91: "from terrestrial ecosystems in the Southern Alps" may be a bit generous if you assume that your 60-km² study area is thus representative.

We have now modified this state to read 'terrestrial ecosystems in the western range front of the Southern Alps' to be more specific about the setting our data represent, but note that in the discussion between lines 213 and 241 we build an argument that our data may be representative of catchments more broadly of the western flank of the Southern Alps.

94: Figure 1: The isobaths are nice, though reveal little about the overall lake depths. Consider adding a maximum depth maybe. Explain why the northeastern parts of Lake Paringa have no catchment area.

We have made the change.

104: "record" -> "records".

We have made the change.

109: "are" -> "were".

Changed to 'there were'

111: "storm-driven landsliding" - How do you know that it was storms and landslides that sourced those sediments?

The storm triggering is inferred and we now acknowledge this on lines 108 and 109. The arguments for these inferences has been documented previously in Howarth et al. (2014), and we do not repeat them here, instead relying on the citation.

Howarth, J. D., Fitzsimons, S. J., Norris, R. J., and Jacobsen, G. E., 2014, Lake sediments record high intensity shaking that provides insight into the location and rupture length of large earthquakes on the Alpine Fault, New Zealand: Earth and Planetary Science Letters, v. 403, p. 340-351.

116: "2SE" - Spell out abbreviation the first time you use it; check grammar in this sentence.

We have made the change.

117: "over the last 1023-1066 years" refers to which calendar year?

We have made the change.

117: "6% of the total mass" - How come these percentages have no error bars?

Uncertainties have now been added to the percentages.

118: Co-, post-, and interseismic phases may need a definition here.

These have now been defined above when the process explanation for the deposits that form these phases is reviewed between lines 101 and 108.

124: Figure 2 would be stronger if it included controls on the absolute ages (14C samples, etc.), although I presume that most of the information presented here is contained in your previously published work (refs. 23, 25).

We refer the reviewer to Figure S2 and S3 where we have 14C dates from multiple cores that confirm the correlations.

130: "1717 CE earthquake cycle" - How can one year be a cycle?

This now reads 'earthquake cycle between the 1717CE earthquake and 1950' to make the meaning clearer.

136: "SS" - How much bedload would those lakes be capable of trapping? Perhaps comment more on this fraction.

Because our reconstruction is based on lakes sediment cores that have grainsizes that are routinely transported as suspended sediment load i.e. <500 microns we do not reconstruct bedload here.

136: "phases" -> "deposits"?

We have made the change.

140: "df=1, t=25, p<0.05" - These and other test statistics are useful only if you describe or mention at least the underlying test.

We have made the change.

146: "from" -> "into"?

We have made the change.

149: "most likely caused by storm-induced landsliding" - see comment on line 111. Given the amount of rain in the western Southern Alps, I find it curious to see so little evidence of storm-induced landsliding in your lake records.

Many of the fine-scale layers of terrestrial sediment in interseismic periods relate to moderate magnitude storms. Here we only capture the very largest events that drive substantial terrestrial landsliding and prolonged landscape responses.

158: “precision” - Yes, but what about accuracy?

Our previous work has established the accuracy of our age-depth modelling approach, e.g., Howarth et al. (2013, 2014, 2021). This approach is outlined in the methods, and we do not repeat it here to keep the manuscript concise.

161: “95% Highest Probability Density Function” - Do you mean a 95% highest density interval?

The terminology is correct and comes from Bronk Ramsey et al. (2008), who developed the age-depth modelling approach.

Ramsey, C. B., 2008, Deposition models for chronological records: *Quaternary Science Reviews*, v. 27, no. 1–2, p. 42-60.

While I welcome the use of intervals, this approach is somewhat at odds with that of the more limited use of statistical testing for differences in the means (e.g. line 140).

The reviewer raises a fair point, but we use Chi-Squared tests here to determine whether or not there is a difference between the distributions for our parameters produced by the MCMC analysis used to propagate uncertainty after Bronk Ramsey et al. (1995).

Ramsey, C. B., 1995, Radiocarbon calibration and analysis of stratigraphy: The OxCal program: *Radiocarbon*, v. 37, no. 2, p. 425-430.

163: “then decayed to values within uncertainty” - Meaning the estimates are then indistinguishable with 95% probability? Please elaborate.

This now reads ‘that were within the 95% HPDF range of the inter-seismic yields’ to improve clarity.

168: Figure 3 should show errors on the calendar ages.

We have made the change.

181: “This demonstrates that lake stratigraphic records can preserve quantitative records of yields from mountain catchments that bridge the gap between instrumental and millennial timescales.” - Agreed, but hardly anything novel?

We respectfully disagree because this is the only example that we are aware of where volumetric reconstructions of lakes have sufficient resolution to resolve quantitative sediment and organic carbon yields associated with discrete extreme events, such as earthquakes and storms.

179: “agree well with independent denudation rate estimates from cosmogenic nuclides” - It might be useful to add these estimates to Figure 3.

We have not done so because there is no OC_{bio} yield data for these independent sediment yield estimates, which means they cannot be plotted on the byplots.

195: “shows” -> “show”.

The sentence has been restructured, so this change is no longer relevant.

201: “extrapolating yields derived from short-term instrumental records over longer timeframes if they are representative of mountain catchments more generally” - Comes back to my initial point about the introductory section about timescales.

We now have a new section titled ‘Implications for extrapolating short term yields over millennial timescales’ between lines 163 and 199 that deals with this issue specifically.

206: “Hence, we hypothesize that extreme event driven landsliding may play an important role in driving transients of SS, OC_{bio} and OC_{petro} yield from catchments” - Perhaps “hypothesize” is the wrong phrase here, as you cite numerous studies with the necessary evidence.

We respectfully disagree. Previous studies have demonstrated that suspended sediment and OC_{bio} yields increase following individual events, but none have quantified the importance of sequential events over century to millennial timescales. For OC_{bio} and OC_{petro}, previous studies have only resolved the initial part of the response to earthquake-driven landsliding.

208: “It also explains why SS [...] from the 32,000 km² Narayani R. catchment draining the Nepalese Himalaya” - Undue breaks in logical flow line the rest of this paragraph: you begin by “explaining” by referring to the previous hypothesis? Without any context, the example of the Nepalese river comes out of the blue and may be difficult to understand without any further details.

We thank the reviewer for this comment and have restructured this section to avoid the breaks in narrative of the argument. See new text between lines 213 and 241.

212: “typhoon” - Do you mean “monsoon”?

Thanks you for spotting this type. It has now been corrected.

213-216: Reads more like introductory material. If this is meant to generalise your findings beyond your study area, you may wish to highlight also the key differences (instead of just some similarities) to other catchments of this size in Oceania.

Based on advice from reviewer 1 we have restructured the text in this section to provide more detail.

217: “Together, this supports our hypothesis” - See comment on line 208. You seem to state a hypothesis and use this to confirm observations to then support your hypothesis. Avoid this circular

reasoning and perhaps rephrase to: “Overall, our data support the hypothesis that extreme event driven landsliding causes substantial transient increases in SS and OCbio export that stand out in sedimentary archives formed over centuries to millennia.”

Thank you for spotting. The section has been rewritten to remove the apparent circular reasoning.

225: “biome-based classification of vegetation and soil carbon stocks” needs a supporting reference.

Apologies – it is now provided.

227: Delete “annual”; it is part of the unit yield at the end of the sentence.

We have made the change.

228: “yield” should be “flux”? At least, you use this term in the following sentence.

We have made the change.

228: “the estimate is uncertain because events may be missing from the EQIL database” - and because the depth of erosion and local OC concentrations are probably even more uncertain.

We disagree – while this would be relevant for the volume of sediment, or supply of rock-derived OC, the mobilisation of most soil requires only very shallow landsliding. These landslides in the database are from satellite mapping where scars are visible, suggesting all soil has been removed.

231: “ Despite the limitations we conclude” - You may want to offer some objective error estimates before concluding.

We have described the data underlying the estimate and the assumptions here. It is difficult to do a more robust uncertainty analysis with this sort of order of magnitude estimate. As such, we have left this as it is.

238: “opportunity to assess whether climate variability is linked to sediment and carbon transfers” - This would also be a good opportunity to frame a hypothesis before interpreting the data. What do you mean by “transfer”: the following sentence refers to “export”?

We have now reworded the sentence to more explicitly state the hypothesis we are testing with the reconstruction data (see lines 265 and 267)

239: “significant difference” - Might need a statistical treatment. Sediment fluxes in Figure 4b seem indistinguishable within error in the first (from the right; at around 1000 CE) and last phase shown.

We agree with the reviewer that statements, such as ‘significant difference’ require a statistical test. In our case we use a Chi-squared test to evaluate whether or not there is a significant difference

between the distributions of sediment and OC_{bio} yields during inter-seismic phases (cf. Bronk Ramsey, 1995). The test reveals that there is a significant difference between these yields and this is reported in the results section above.

Yet, the Fiord Precip Index (Figure 4d) differs in that it steadily increased in the last phase, while it remained roughly constant in the first.

The low temporal resolution of the reconstructed yields means that only a first order comparison with the proxy climate reconstructions is appropriate. Figure 4 clearly shows that the reconstructed yields are higher in phase 1 and 4, when precipitation indexes are generally higher, and the climate indices controlling precipitation at the site are optimally aligned i.e. SAM and SOI are in phase and negative.

Similarly, the second and third phase of biospheric OC yields (Figure 4a) seem to be indistinguishable from each other, whereas the SOI (red line, Figure 4f) was mostly negative during the second, and mostly positive during the third phase. Some more formal time-series analysis may be helpful here.

It is not appropriate to compare the yields to SOI alone because precipitation in the study region is driven by the interaction between SOI and SAM. As we state, between lines 273 and 293, precipitation is highest when SOI and SAM are in phase (have the same trend) and their absolute values are negative. Figure 4 shows that these time periods overlap with when our reconstructed yields are highest.

240: "In Lake Mapourika, the shorter length of the record precludes robust comparison" - Unclear. How much shorter is the record and why is this still sufficient then for drawing conclusions about differences in sediment and OC yields?

We only have an OC_{bio} yield estimate for the seismic cycle associated with the 1717 CE earthquake, which means there is an n of one interseismic phase for this lake. Consequently, we cannot make any meaningful comparison to the paleoclimate data.

246-248: Errors needs specification; please note that the errors of low and high yield phases overlap. How well can you distinguish these mean rates from each other?

Thanks for spotting this. We now report what the error estimates refer to. The subsequent statement is incorrect, the 1717 CE interseismic yields are higher than the other interseismic yields when uncertainties are considered.

249: "This suggests that runoff" - This proposed link may need elaboration. How does SHWW connect to precipitation, and how does this connect to runoff in your study area?

Thanks for pointing out the need for clarification. Detailed information on the links between SHWW, precipitation, and runoff are available in the citations provided. However, we have modified the text between lines 273 and 293, so there is less ambiguity in the relationship between precipitation, SHWW and the SAM and SOI climate indices.

250: “during time periods between large earthquakes and storms” - Why “between”? Should those climatic fluctuations not be superimposed onto the earthquake cycle?

We argue that the landscape responses to earthquake-induced landsliding are superimposed on the more secular changes in yields driven by climate. The high amplitude of the response to earthquake-induced landsliding means that during post-earthquake increases in sediment yield, it is not possible to evaluate how the catchments respond to climate.

251: “confirm hypotheses from instrumental studies that use spatial variations in climate to show runoff controls OC_{bio} export” - This statement is a bit vague. Please be more specific.

The statement has been modified to make it explicit that we are referring to hypotheses derived from space for time substitution experiments based on spatial gradients in climate and short term yield estimates from river gauging (lines 291 and 293).

254: “Fiord” Precipitation Index needs explanation.

Citations are provided to the relevant literature where detailed explanations of the proxies are documented. We do not think it is appropriate to provide detailed explanations of how previously published proxy data series were reconstructed, particularly given the relatively short format of the article.

268: “fit to empirical data from mountain river catchments globally” - Is this fit thus prone to the limitations that you cautioned against earlier, i.e. interpolating short-term measurements?

We agree with the reviewer on this point. In fact it is the main point we are trying to make. We show that a reconstructed time series of hydroclimate and OC_{bio} yields demonstrate the same relationship as derived from space for time substitution experiments based on short-term yield data from rivers. We have restructured this entire section in an effort to make this clearer.

270: “described by a function of” -> “estimated from”. It might pay off to add that the equation in line 272 estimates only the average river particulate biospheric OC concentration.

We have now articulated how we get from a concentration to a yield in lines x and y.

273: “pre-factor that is related to the slope angle of the catchment” - Unclear. Why is this coefficient only related to slope?

Arguments for this are outlined in the original publication, Hilton (2017), which we cite. It is beyond the scope of our manuscript to relitigate this published interpretation.

Hilton, R. G., 2017, Climate regulates the erosional carbon export from the terrestrial biosphere: *Geomorphology*, v. 277, p. 118-132.

274: “Assuming $\alpha = 0.052$ ” - Alpha is not dimensionless; please add units. Both alpha and gamma have errors, which should enter the following estimates.

We thank the reviewer for pointing this out. We have now added those units for Alpha where it is reported. The following discussion uses a single parameterisation of the model based on the discussion in the text, and so the quoted uncertainty provided in the original empirical fit of the data is not relevant in this case.

276: Explain “CI”.

We have made this change.

281: “measured” -> “inferred”.

We have made this change.

283: “can be explained by the erosion model” - Do you mean the equation in line 272? Why do you require the data to fit the model? If anything, it should be the other way ‘round. You should contemplate that the simple model in line 272 may fail to capture all variance in the data. Hysteresis would be a first effect to consider when looking at relationships between runoff, discharge, and material concentrations in rivers.

See the response to the major comment above.

295: “doesn’t” -> “does not”; “for the predicted increase in severity of the largest flow events ... because it scales non-linearly with runoff” - Do you mean that equation in line 272 is non-linear? But why then are the largest flow events a problem? Either the model is capable of accurate prediction or it is not. Note that you refer to “erosion” in line 296, whereas the equation estimates concentrations.

We use mean annual runoff in our first-order analysis because this is the simplest way to account for modelled changes in runoff due to anthropogenic climate change. As such, the modelled OC_{bio} could be underestimated because forecasts predict higher runoff extremes over the next century. Hence, we acknowledge the conservative nature of our estimate.

289: “Building on this assessment of the lake records using the runoff-driven model” - This reads as if you assess the data based on your model, whereas the way that statistical inference works is vice versa.

Thanks for pointing out how this read, as it was not our intended meaning. We have reworded the sentence to capture that we are building on the observed relationship between reconstructed OC_{bio} yields and hydroclimate, combined with the agreement between reconstructed and modelled yields (see lines 313 and 315).

304: “how orogenesis, erosion and weathering contribute to the carbon cycle” - Your study addresses none of these likely contributors directly.

This statement is taken out of context. It is part of a sentence describing the general processes that control the geological carbon cycle. The paragraph then goes on to outline the specific contribution our study makes to these processes.

310: “fluxes are mediated by hydroclimatic variability” - What do you mean by “mediated” here? Please see comments on line 239.

See response to comments on line 239.

312: “climate change from the Little Ice Age” - Please remind readers about the timing of this in the Southern hemisphere, and consider showing this interval in Figure 4.

This change has been made.

314: “Our data shows that erosion of OCbio could act as a negative feedback in the Earth System” - Maybe tone this down a bit. You generalise your findings from sediments in two lakes in New Zealand to erosion in the mountain catchments upstream, and then the Earth System? Maybe remind readers which data or evidence you refer to here.

We have modified the sentence to that it states that our reconstructions support the hypothesis based on global scale space for time substitution experiments on modern observations that erosion of OCbio could act as a negative feedback in the Earth System (see lines x and y).

315: “which” refers to?

The sentence has been modified and the comment is no longer relevant.

317 “climates associated with warmth can enhance erosion of OCbio from terrestrial ecosystems, promote its transfer into long-term sedimentary storage, and drawdown CO₂ from the atmosphere (Fig. 4)” - Not sure whether Fig. 4 shows this clearly, let alone whether it is prudent to generalise your findings; see previous comment.

See response to comments on line 239.

321: “is thought to be typical” sounds vague and as if it reflects models fit to data?

We do not understand the comment as this sentence refers to the published ratio between runoff and carbon drawdown fluxes for silicate weathering.

480-494: Please explain the nature of all errors cited. This consistency is essential as you base a lot of your conclusions on quantitative estimates, in which these errors will accumulate. Some errors

seem to be standard deviations about the mean, other confidence intervals, and yet others credible intervals. If combining these, please consider their differing interpretations.

We do not understand the relevance of the comment because none of the uncertainties in this paragraph are used in our calculations. However, we do now report the type of uncertainties reported by the original authors.

517: "sill that separates the Windbag Basin from the Hall Basin" - First time you mention the Hall Basin without much context.

541: "derive" -> "estimate". The durations of these phases thus should have age errors.

The reviewer is correct. The durations have uncertainties, which are propagated through our sediment and OC_{bio} yield analyses. We now state this overtly on line 419.

554: Index "n" is the number of voxels?

Correct, index n is the number of voxels.

564: "$\pm 5\%$" makes no sense mathematically.

Thanks for pointing this out this typo. We have reworded the sentences between lines 441 and 444 to make sure the meaning is clear.

564: "uncertainty was propagated through" - How?

Thanks for picking up this omission. This now reads 'a conservative uncertainty for the W of each phase of $\pm 5\%$ (2SE) was adopted and propagated in quadrature through \forall calculations' to show how uncertainty was propagated.

568: "A is the catchment area with hillslope angle above 15°" - I understand that you use this threshold to include landslide-prone hillslopes only. Why would landslides not occur beyond this threshold? Debris flows might enter the lake(s) at lower terrain inclinations. You might unnecessarily bias your sediment and OC yields by assuming that only landslide input is relevant. It may be a fair assumption that soil erosion, surface runoff, etc. produce negligible input, but you cannot infer the relevance of landslides from your data if you admit little else in terms of other geomorphic processes.

See the response to the main comment above.

615: "from two range front catchments near Lake Mapourika with the same physiographic characteristics as Potters Creek" - This reads a bit vague given the previous efforts that have tried to link denudation rates from cosmogenic nuclides to all sorts of terrain characteristic. Do these

sampled catchments also contain parts of the foreland? Maybe show these sample locations in Fig. 1?

Fair point, we now specifically state what physiographic metrics are similar between the catchments i.e. morphometry, rock type, vegetation and climate on line 502.

624: “similar physiographic setting (New Zealand and Taiwan)” - See previous comment about similar physiographic setting. Maybe limit this to data from New Zealand?

There are so little data on landslide grain size distributions, so we opted to keep the global dataset to make the estimate conservative.

629: “there is no comparable data for validating the Lake Paringa SS yields” - Assuming that the rates derived from cosmogenic nuclide concentrations reflect the true rates? It might be sufficient to compile the available data on erosion and denudation here: each of the methods has its drawbacks, but they constrain the range of rates quite well.

Erosion and uplift rates vary along the Southern Alps (Norris and Cooper, 2007; Hovius et al., 1997), and there are no ^{10}Be -derived estimates of denudate as far south as Lake Paringa. Consequently, we limit our comparison between the lake-derived and ^{10}Be -derived yields to catchments with comparable physiography.

Hovius, N., Stark, C. P., and Allen, P. A., 1997, Sediment flux from a mountain belt derived by landslide mapping: *Geology*, v. 25, no. 3, p. 231-234.

Norris, R. J., and Cooper, A. F., 2007, The Alpine Fault, New Zealand: surface geology and field relationships: *A Continental Plate Boundary: Tectonics at South Island, New Zealand*, p. 157-175.

636: “Our estimates of OCbio yields for a given SS yield agree well with global values” - What do you mean by “agree well”?

We have modified the text between lines 519 and 526 to be more specific. It now reads ‘...scale with global estimates of these yields derived from river gauging.

640: “This trend is likely compounded” - Which trend? Do you mean “confounded” or “compounded”? If you mean the latter, what are the other components other than high OC stocks in soils?

The text has been reworded so the meaning is now clearer (see lines 519 and 526).

646: “Seal et al.,33” - Reference appears to be incomplete in the list. Please check.

Thanks for picking up this error it has been corrected.

651: “window” -> “interval”?

This change has been made.

651: What is the “typical return interval of earthquakes”?

This is stated explicitly on lines 88 and 358.

658: “assess” - “use”?

Assess is the correct terminology in our view because the biome estimates are used to estimate the local carbon stocks for each earthquake.

661: “may represent over- or under- estimates” -> “be misestimates”. So why then would the sum of these be any more accurate than individual events? Please elaborate.

Because when sampling a relatively large number of events under and overestimates are more likely to cancel each other out.

Fig. S7: Caption should explain dotted lines and green box.

These are now specified in the caption.

Fig. S8: Regression equation is valid only for mean F_{petro} . How do you deal with the variance when you use this model for predictions?

As stated in the methods, we propagate an uncertainty of 2 standard errors of the predicted value through our calculations to capture uncertainty that derives from the spread of D_{13C} and F_{petro} values.

Fig. S9: Please explain the two lines. Why do the global data have no errors?

In response to the comments from reviewer 1 we have provided the equations from the global regressions that derive from Galy et al (2015). These authors do not report uncertainty on their OC_{bio} or OC_{petro} estimates so we cannot plot them.

27 April, 2025

To Reviewer 2,

We thank Reviewer 2 for the careful re-review of our manuscript COMMSENV-24-2345A: “Long-term carbon export from mountain forests driven by hydroclimate and extreme event-driven landsliding”. Your insightful comments have once again helped us improve the clarity of the manuscript. We have made substantial corrections to the manuscript or provided in-depth clarification and rebuttal in our response below (blue text).

Kind regards,

J. Howarth

Reviewer #2 (Remarks to the Author):

Review of manuscript #COMMSENV-24-2345A: “Long-term carbon export from mountain forests driven by hydroclimate and extreme event-driven landsliding” submitted by J. Howarth and colleagues

Dear Authors,

Thank you for your detailed reply letter concerning the remarks and suggestions of the previous round of reviews. Your answers have clarified most of my initial queries, and I appreciate how thoroughly you revised the original manuscript. Below I list a number of comments that you may want to look at once more. These refer mainly to how you handle errors and how this might affect your discussion about the global relevance of your study. Other than that, thanks again for this sneak preview: I have learned a lot about how one can estimate sediment and OC yields from the western Southern Alps.

We thank the reviewer for their detailed and thoughtful comments, which helped improve the manuscript substantially. We are glad that the reviewer recognises the effort and extent of the changes we made.

Line-specific comments (refer to track changes version):

2: “extreme event-driven landsliding” – I maintain my original comment that you show very little evidence of landslides.

As mentioned previously, the argument that hyperpocynite stacks represent pulses of increased sediment flux from landsliding has been established in previous publications (Howarth et al., 2012; Frith et al., 2018; Wang et al., 2020). The critical evidence that supports a landsliding mechanism derives from the organic geochemistry of these deposits that shows the sediments are of terrestrial

origin (Howarth et al., 2012; Frith et al. 2018), derived from high elevation hillslopes where landsliding is the dominant erosion process (Wang et al., 2020; Hovius et al., 1997), and that these sediments are generally mobilised from deeper soils implying landsliding is the mobilisation mechanism (Wang et al., 2020). Given this previous work establishing the importance of landsliding, we prefer to leave the title unchanged.

Frith, N. V. et al. Carbon export from mountain forests enhanced by earthquake-triggered landslides over millennia. *Nature Geoscience* (2018).

Hovius, N., Stark, C. P. & Allen, P. A. Sediment flux from a mountain belt derived by landslide mapping. *Geology* 25, 231-234 (1997).

Howarth, J. D., Fitzsimons, S. J., Norris, R. J. & Jacobsen, G. E. Lake sediments record cycles of sediment flux driven by large earthquakes on the Alpine fault, New Zealand. *Geology* 40, 1091-1094 (2012).

Wang, J. et al. Long-term patterns of hillslope erosion by earthquake-induced landslides shape mountain landscapes. *Science advances* 6, eaaz6446 (2020).

70: “has been revealed using spatial regression models” – Consider replacing “revealed” by “estimated” for reasons that you spell out in this new paragraph. Again, I have yet to see “spatial” regression models. The “space for time” substitution you mention in line 78 is not necessarily a spatial regression model in the geostatistical sense. Try avoiding this ambiguity.

To avoid ambiguity, we have removed mention of spatial regression models in the paragraph starting on line 68 in the revised version of the text.

79: “scaling between variables spans an order of magnitude” – Please specify these variables. This correction has been made on line 77 of the revised text.

171: How do you define or estimate the “long-term average” and why? You put a lot of emphasis on the importance of decadal to millennial intervals in sedimentary records, and hence might also want to establish what you feel is the necessary background for reference.

We agree this is important to establish. To avoid any ambiguity about how we define ‘long-term’ for the purposes of this paper we define it explicitly in the first sentence of the paragraph starting line 165 in the revised manuscript i.e. ‘...assess how yields deviate from the long-term average, defined here as the average yields over the record duration.’

173: “average SS yield over the record duration was ...” - See previous comment: is this the “long-term average”?

This is correct and is now defined explicitly in the first sentence of the paragraph to avoid any ambiguity (see above response).

190: What are the “timescales relevant to the geological carbon cycle” and how can you tell.

This refers to the reservoirs and fluxes which govern the net exchange of carbon between the atmosphere, biosphere and ocean system and the lithosphere (see Hilton and West, 2020, for a review and other primary literature on this topic). The timescales come from considering the sizes of these reservoirs (~600 PgC, ~3000 PgC and ~38,000 PgC respectively for the atmosphere, biosphere and oceans) and the main fluxes into and out of the lithosphere via volcanic degassing, silicate weathering, organic carbon burial and rock organic carbon weathering, which vary but are

approximately 0.1 PgC/yr. Given this flux and the reservoir sizes, one can define a timescale over which these fluxes are relevant as the residence time = mass of reservoir / fluxes. For the ocean+atmosphere+biosphere and these exchange fluxes, the timescale is ~100,000 years (see Figure 2 in Hilton and West, 2020).

We have not expanded text in the manuscript on this point, as it is a relatively common statement to link to the geological carbon cycle. But on reflection, it is simpler to make this sentence self-contained “We therefore, recommend caution when extrapolating sediment and carbon yields calculated over years to a decade from river gauging data to longer timescales”

Hilton, R. G. & West, A. J. Mountains, erosion and the carbon cycle. *Nature Reviews Earth & Environment* 1, 284-299 (2020).

193: “Our approach to reconstructing sediment and carbon yields from lake sediments could be deployed globally” – How would that work? It seems quite an effort to study what you did for a couple of lakes.

We are pleased that the reviewer acknowledges the considerable effort required to reconstruct our quantitative sediment yields. Our argument is that our approach could be deployed to assess how extreme events influence sediment and carbon yields in a broader range of physiographic settings. We acknowledge that it would be impractical to quantify similar fluxes from every suitable lake globally. Hence, we have removed the term ‘globally’ from the sentence starting line 187 in the revised manuscript to avoid ambiguity. However, the importance of the original points still stands because our approach provides one of the only means of quantitatively assessing how extreme events derive temporal variations in sediment and carbon yields.

194: “over timeframes relevant to the carbon cycle” – See previous comment.
See the response to the previous comment.

197: “ours” → “our”.

The typo has been corrected.

234: “These small, highstanding, and forested catchments have OC_bio yields that are, on average, four times higher than yields from other physiographic settings” – Does this comparison rely on short-term yields that you caution extrapolating with each other?

The reviewer is correct; all current influences on global organic carbon yields are derived from short-term river gauges. But here we are simply making a point for their global relevance. To avoid overly complicating the text here, we decided not to mention the fact that these modern day yields are likely to be an underestimate.

237: “our findings, while based on only two catchments, may have global relevance because ... SS and OC_bio yields derived over years to a decade from river gauging data may underestimate long-term yields for catchments that are a significant source of OC_bio export to the global ocean” – Suggest revisiting this statement for two reasons. (1) Other studies in similar settings might differ concerning their earthquake cycles or length of record, and accordingly so the mismatch between shorter- and long-term rate estimates. It is a valid point, though, to stress the discrepancy between gauge readings and volumetric estimates of lake fills. (2) How are your catchments “significant source of OC_bio export to the global ocean” with much of their output trapped in lakes?

Regarding the first point raised by the reviewer, while we agree that the length of the earthquake cycle will vary regionally, as do the length of existing river gauging records, earthquake-generated OC_{bio} yields have only been captured in river gauging records in a few catchments globally. Therefore, very few of the records used to quantify OC_{bio} export to the global ocean capture these extreme events.

Concerning point two, we are using the lake sites as long-term gauging stations for river yields, so the fact they trap all the sediment is an asset in the context of our study. The vast majority of small mountain river catchments do not drain into lakes prior to discharging into the ocean. In the paragraph leading to the statement about the broader significance of our work, we emphasise the similarities between our catchments and other high-standing forested catchments in Oceania that are important sources of OC_{bio} export to the global Ocean. Given the strong similarities, we think it useful to point out the potential broader implications of our work. To avoid overstating the significance, we have replaced 'global relevance' with 'have broader implications' and made sure we specifically mention that the analogy between our study catchments and those more generally in Oceania is the reason why we think current OC_{bio} export to the global ocean may be underestimated (see sentence beginning line 229 in the new manuscript).

238: "suggests" → "suggest".
The typo has been corrected.

242: "the spatial scale of event-driven landsliding is low relative to the total catchment area, limiting the magnitude of transient increases in OC_{bio} export caused by extreme-event driven landsliding" – This may need a supporting reference or two. One could think that, in these settings, the mismatch between short-term readings and long-term effect of rare earthquakes or rainstorms could be even higher.

We have rewritten the text between lines 232 and 239 to elaborate more clearly on the logic behind this point, supporting it with citations. It now reads: 'In these settings, where catchment areas exceed $\sim 10^4$ km², the spatial extent of event-driven landsliding is often less than the total catchment area (e.g. ref¹⁸). If the ratio of disturbed catchment area to the total catchment area controls the magnitude of transient increases in yield⁴⁰, then transient increases in OC_{bio} export caused by extreme-event driven landsliding may not be as pronounced'.

247: "showed a negligible increase in SS or OC_{bio} yields following the 2016 Mw7.8 Gorka earthquake that triggered extensive landsliding" – Perhaps it would be useful to elaborate briefly why. Before the background of discussing a more regional, if not global, relevance, this particular study (ref. 18) seems to limit the notion of wider applicability of yours. Quite the contrary, this difference motivates a closer look at the underlying controls and promises some interesting future work.

We have now acknowledged the need for future work with a sentence starting on line 245.

249: "only impacted the eastern part of the catchment and represented only a modest increase over annual monsoon driven rates of landsliding" – And this scenario of localized landsliding is something that you can exclude for the western Southern Alps? My understanding is that the Gorkha earthquakes still triggered tens of thousands of landslides.

Catchments in the Southern Alps are an order of magnitude smaller, and rupture of the range-bounding Alpine Fault in Mw7.5-8.0 earthquakes ruptures 200-400 km of range bounding fault length. Models show that these earthquakes cause dense landsliding across the full extent of the

range front catchments of the South Alp's western flank (see Croissant et al., 2019).

261: "while temporal variations in EQIL rate may not be captured by the 50-year timeframe" – I am not even convinced that the estimated mean EQIL rate is reliable, given that the catalogue you use is heavily biased by recent, and more detailed, studies.

There is no doubt that the existing published EQIL database has its limitations, and we acknowledge the point made by the reviewer in the manuscript text (see lines 256 to 258). However, we note that any mapping bias due to improvements in inventory quality and extent for more recent events would bias the EQIL rates we report towards underestimates, making our estimate conservative. Even this conservative estimate is equivalent to the OC_{bio} flux from the Ganges – Brahmaputra, one of the largest sources of OC_{bio} to the global Ocean. Hence, we feel that the analysis is a valid and useful discussion point and have retained it.

276: "to assess the hypothesis derived from space for time substitution experiments that temporal variability in hydroclimate drives changes in sediment and carbon yields" – Note that the space-for-time approach is a hypothesis on its own. To test whether variable hydroclimate is a possible driver, you may need to demonstrate that sediment and OC yields remain unchanged without considering hydroclimate.

Unfortunately, we do not fully understand the point the reviewer is trying to make here. However, while we agree that the space-for-time approach has been used to hypothesise that temporal variability in hydroclimate drives changes in sediment and carbon yields. Based on our reading, this is exactly what the sentence between lines 271 and 273 says. Further, it is not clear to us why it is necessary to demonstrate that sediment and OC yields remain unchanged despite hydroclimate if we are trying to test the hypothesis that sediment and OC yields are driven by hydroclimate. To the contrary, we argue that the fact that our reconstructed sediment and OC yields scale with hydroclimate variation through time and provides novel time series data that supports the hypothesis based on space-for-time substitution. On this basis we have left the text unchanged.

303: "our findings also confirm hypotheses from space for time substitution experiments that use OC_{bio} yield from river gauging and spatial gradients in climate" – Not sure whether you can "confirm" these hypotheses at all: they would be fact eventually.

We thank the reviewer for suggesting we modify this statement. We have changed 'confirm' to 'support' in line with the concept that it is only possible to falsify a hypothesis and not confirm that it is true.

How is the space-for-time substitution for gauging data not prone to a possible short-term bias that you cautioned against earlier?

It absolutely is prone to these issues; that is why our hypothesis test using a reconstructed quantitative time series of these fluxes that spans longer timeframes is so important.

316: Coefficient alpha has incorrect unit: this must also depend on exponent gamma. If the equation in line 315 is a regression model, it misses an error term. At the very least, the OC yields in this and the following paragraph should carry the estimated regression error.

We now give the units of alpha as $mg L^{-1} mm^{-1} \gamma day^{\gamma}$ we hope this clarification is useful for future readers.

We thank the reviewer for pointing out that we should propagate the uncertainty from the regression through the calculation of OC_{bio} yields from the model. We have now propagated the 95% CI from the regression exponent through the calculations. Doing so increases the uncertainty of our estimates but doesn't change any of the conclusions. The estimates with revised uncertainty appear in the text between lines 312 and 326. The units for alpha are correct, see (Hilton et al. 2019).

318: “we assume $\alpha = 0.052$ and $\gamma = 1.37$, which reflects the Lake Paringa catchment” – Why assume? Do you not obtain this via regression? And what about the catchment feeding Lake Mapourika?

We thank the reviewer for pointing out the lack of clarity in this sentence and have re-written it to improve clarity. The reviewer is correct; the exponent is derived from the regression of Hilton (2019). We now mention this explicitly in line 307, including uncertainty. We assume an alpha of 0.052 based on the arguments of Hilton (2019), who shows that alpha scales with catchment slope. We choose an intermediate value, which is consistent with the Lake Paringa catchment that has some areas of moderate slope, as stated on line 309.

As stated on line 277, we do not model OC_{bio} yields for Lake Mapourika because the reconstructed time series of yields from that lake is too short to make meaningful comparisons with the reconstructed hydroclimate proxies.

320: “mean annual runoff of 3848 ± 440 (95% CI) calculated from the Lake Paringa catchment rain gauge” – Runoff needs unit. How long is this record and how does it safeguard against the problem of unduly extrapolating over longer time periods?

The units of $mm\ yr^{-1}$ have been added on line 305 of the revised text. The mean annual runoff record has data from 1956 to 2008. As it spans over 50 years we think it is appropriate for a first order comparison to the youngest epoch of reconstructed yields.

356: “Our reconstructions support the hypothesis based on global scale space for time substitution experiments on modern observations” – See previous comments. I would be more careful in discussing how well you can extrapolate your local findings for reasons that you clearly spell out.

We respectfully disagree. Our longer term reconstruction of sediment and OC_{bio} yields scale with hydroclimate variation over the last millennium. This record provides some of the first evidence that the hypothesised relationship between OC_{bio} yields and hydroclimate, derived from the spatial correlation of these variables globally, plays out over time scales that approach those relevant for the geological carbon cycle. Given this, we think it is appropriate to point out the broader implications of a hydroclimate control on OC_{bio} export, which we do in the final paragraph of the paper.

912/Fig. 4: It might be useful to add to the boxplots of SSY and OC which of the four intervals you can tell apart from each other within error. In other words, how well can you establish that yields in phases 1 and 4 of SSY and phases 2 and 3 of OC are different?

We have already plotted the median and 95% HPDF range on the figure. We now make what is shown in the figure clear in the caption. We have also added Bonferroni-corrected pairwise chi-squared tests to demonstrate differences in SS and OC_{bio} yields between the phases. The methods have been updated (see lines 501 and 503), the results are shown as letters above the yield distributions in figure 4 a and b, and the caption to the figure has been updated to describe what

this shows. The results confirm that the 1717 CE phase yields are different to 1400 CE for both SS and OC_{bio}.